# A K$^+$-selective CNG channel orchestrates Ca$^{2+}$ signalling in zebrafish sperm

**Sylvia Fechner[1†], Luis Alvarez[1], Wolfgang Bönigk[1], Astrid Müller[1], Thomas K Berger[1], Rene Pascal[1], Christian Trötschel[2], Ansgar Poetsch[2], Gabriel Stölting[3], Kellee R Siegfried[4], Elisabeth Kremmer[5], Reinhard Seifert[1], U Benjamin Kaupp[1]***

[1]Abteilung Molekulare Neurosensorik, Center of Advanced European Studies and Research, Bonn, Germany; [2]Lehrstuhl Biochemie der Pflanzen, Ruhr-Universität Bochum, Bochum, Germany; [3]Institute of Complex Systems 4, Forschungszentrum Jülich, Jülich, Germany; [4]Biology Department, University of Massachusetts Boston, Boston, United States; [5]Institut für Molekulare Immunologie, Helmholtz-Zentrum München, München, Germany

*For correspondence: U.B. Kaupp@caesar.de

Present address: †Department of Molecular and Cellular Physiology, Stanford University, Stanford, United states

**Abstract** Calcium in the flagellum controls sperm navigation. In sperm of marine invertebrates and mammals, Ca$^{2+}$ signalling has been intensely studied, whereas for fish little is known. In sea urchin sperm, a cyclic nucleotide-gated K$^+$ channel (CNGK) mediates a cGMP-induced hyperpolarization that evokes Ca$^{2+}$ influx. Here, we identify in sperm of the freshwater fish *Danio rerio* a novel CNGK family member featuring non-canonical properties. It is located in the sperm head rather than the flagellum and is controlled by intracellular pH, but not cyclic nucleotides. Alkalization hyperpolarizes sperm and produces Ca$^{2+}$ entry. Ca$^{2+}$ induces spinning-like swimming, different from swimming of sperm from other species. The "spinning" mode probably guides sperm into the micropyle, a narrow entrance on the surface of fish eggs. A picture is emerging of sperm channel orthologues that employ different activation mechanisms and serve different functions. The channel inventories probably reflect adaptations to species-specific challenges during fertilization.

## Introduction

Fertilization is a complex task that, for different species, happens in entirely different spatial compartments or ionic milieus. In aquatic habitats, gametes are released into the water where sperm acquire motility and navigate to the egg. By contrast, mammalian fertilization happens in confined compartments of the female oviduct. From invertebrates to mammals, sperm use various sensing mechanisms, including chemotaxis, rheotaxis, and thermotaxis, to gather physical or chemical cues to spot the egg. These sensory cues activate various cellular signalling pathways that ultimately control the intracellular Ca$^{2+}$ concentration ([Ca$^{2+}$]$_i$) and, thereby, the flagellar beat and swimming behaviours (*Alvarez et al., 2012*; *Darszon et al., 2008*; *Eisenbach and Giojalas, 2006*; *Florman et al., 2008*; *Guerrero et al., 2010*; *Ho and Suarez, 2001*; *Kaupp et al., 2008*; *Publicover et al., 2008*). In species as phylogenetically distant as sea urchin and mammals, these pathways target a sperm-specific, voltage-dependent Ca$^{2+}$ channel, called CatSper. Signalling events open CatSper by shifting its voltage-dependence to permissive, more negative V$_m$ values. This shift is achieved by different means. In sea urchin sperm, opening of a K$^+$-selective cyclic nucleotide-gated channel (CNGK) causes a transient hyperpolarization (*Bönigk et al., 2009*; *Strünker et al., 2006*); the hyperpolarization activates a sperm-specific Na$^+$/H$^+$ exchanger (sNHE) (*Lee, 1984*, *1985*; *Lee and Garbers, 1986*) resulting in a long-lasting alkalization that shifts the

**eLife digest** Mammalian sperm cells fertilize egg cells inside the female's body; while for fish and other marine animals it is common for fertilization to take place outside in the environment. In general, sperm cells become attracted to egg cells by various chemical or physical signals. Sperm detect these cues and generate electrical signals that control their own movements and eventually guide sperm to the egg.

In 2009, researchers identified a potassium ion channel, called CNGK, that starts the electrical signal in the sperm cells of sea urchins. This channel is activated by signalling molecules inside cells, called 'cyclic nucleotides', and its activity ultimately leads to calcium ions flowing into the sperm cell's tail. This influx of calcium ions in turn controls the beating of the tail and, thereby, steers the sperm cell towards the egg.

It was previously thought that this CNGK channel is found only in animals without a backbone (i.e. in invertebrates). However, Fechner et al. – including some of the researchers involved in the 2009 work – now report that the CNGK channel also exists in the sperm cells of a freshwater fish, the zebrafish. Unexpectedly, the CNGK channel is located in the heads of this fish's sperm cells rather than in the tails.

Electrophysiological experiments then revealed that the fish version of CNGK is not activated by cyclic nucleotides, but is activated when the inside of the cell becomes more alkaline. In zebrafish sperm, a more alkaline pH inside the cell causes calcium ions to flow in and this influx of calcium ions triggers a unique spinning-like swimming movement that is different from the swimming of other sperm from other species. Fechner et al. suspect that this unusual swimming behaviour guides sperm through a small hole in the protective coating of fish eggs, which eventually leads to fertilization.

These findings suggest that while channel proteins found in sperm cells from different species look similar and serve similar roles, they are activated in ways that can be very different.

voltage dependence of CatSper and leads to a $Ca^{2+}$ influx (*Kaupp et al., 2003*; *Seifert et al., 2015*). By contrast, in human sperm, the shift is achieved by direct stimulation of CatSper with prostaglandins and progesterone in the seminal fluid or the oviduct (*Brenker et al., 2012*; *Lishko et al., 2011*; *Smith et al., 2013*; *Strünker et al., 2011*).

Navigation of fish sperm and the underlying signalling pathways must be arguably different. First, teleost fish are lacking CatSper channels (*Cai and Clapham, 2008*), although activation of sperm motility requires $Ca^{2+}$ influx (*Alavi and Cosson, 2006*; *Billard, 1986*; *Cosson et al., 2008*; *Morisawa, 2008*; *Takai and Morisawa, 1995*) stimulated by hyper- or hypoosmotic shock after spawning into seawater or freshwater, respectively (*Alavi and Cosson, 2006*; *Cherr et al., 2008*; *Krasznai et al., 2000*; *Morisawa, 2008*; *Vines et al., 2002*). Therefore, $Ca^{2+}$ signalling in fish sperm must involve molecules different from those in marine invertebrates and mammals.

Second, the ionic milieu seriously constrains ion channel function. Sperm of freshwater fish, marine invertebrates, and mammals are facing entirely different ionic milieus. $K^+$ and $Na^+$ concentrations in freshwater are extremely low (70 µM and 200 µM, respectively) compared to the orders-of-magnitude higher concentrations in seawater or the oviduct (*Alavi and Cosson, 2006*; *Hugentobler et al., 2007*). Furthermore, $[Ca^{2+}]$ in seawater is high (10 mM), whereas in freshwater it is low (< 1 mM). The low salt concentrations in freshwater probably require distinctively different ion channels. In fact, none of the ion channels controlling electrical excitation and $Ca^{2+}$ signalling of fish sperm are known.

Finally, fish sperm are not actively attracted to the whole egg from afar by chemical or physical cues, i.e. chemotaxis, thermotaxis, or rheotaxis (*Cosson et al., 2008*; *Morisawa, 2008*; *Yanagimachi et al., 2013*). Instead, many fishes deposit sperm directly onto the eggs. For fertilization, sperm must search for the narrow entrance to a cone-shaped funnel in the egg coat – the micropyle – that provides access to the egg membrane. Sperm reach the micropyle probably by haptic interactions with tethered molecules that line the egg surface and the opening or interior of the micropyle (*Iwamatsu et al., 1997*; *Ohta and Iwamatsu, 1983*; *Yanagimachi et al., 2013*). At

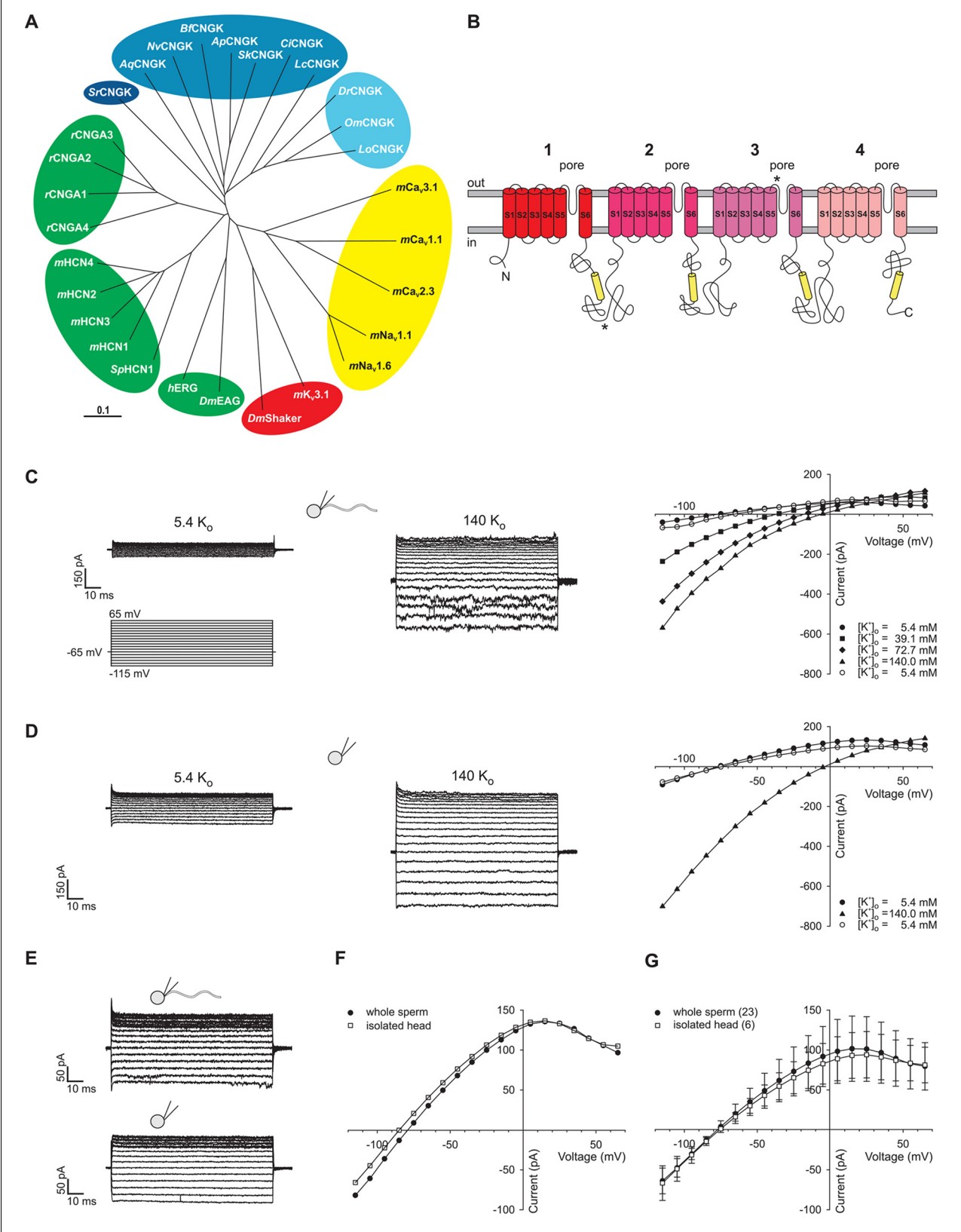

**Figure 1.** Identification of *Dr*CNGK channel homologues and of a K$^+$ channel in *D. rerio* sperm. (**A**) Phylogenetic tree (*Page, 1996*) of various ion channel families. The CNGK channel family exists in protozoa (dark blue), marine invertebrates and fish (medium blue), and freshwater fish (light blue).
*Figure 1 continued on next page*

*Figure 1 continued*

The HCN, CNG, and KCNH channel families are highlighted in green; voltage-gated $Na_v$ and $Ca_v$ channels are highlighted in yellow; and voltage-gated $K_v$ channels are highlighted in red. The following ion channel sequences were used: CNGK channels from zebrafish (*Dr*CNGK), rainbow trout (*Om*CNGK), spotted gar (*Lo*CNGK), West Indian Ocean coelacanth (*Lc*CNGK), sea urchin (*Ap*CNGK), acorn worm (*Sk*CNGK), amphioxus (*Bf*CNGK), starlet sea anemone (*Nv*CNGK), vasa tunicate (*Ci*CNGK), sponge (*Aq*CNGK), choanoflagellate (*Sr*CNGK); murine HCN channel subunits 1 (*m*HCN1), 2 (*m*HCN2), 3 (*m*HCN3), 4 (*m*HCN4), and the HCN channel from sea urchin (*Sp*HCN1); rat CNGA subunits A1 (*r*CNGA1), A2 (*r*CNGA2), A3 (*r*CNGA3), and A4 (*r*CNGA4); the KCNH channels from fruit fly (*Dm*EAG) and human (hERG); murine voltage-gated $Na_v$ (*m*$Na_v$ 1.1 and *m*$Na_v$ 1.6) and $Ca_v$ channels (*m*$Ca_v$1.1, *m*$Ca_v$2.3 and *m*$Ca_v$3.1) and voltage-gated $K_v$ channels from fruit fly (*Dm*Shaker) and mouse (*m*$K_v$3.1). Full-length Latin names and accession numbers are given in experimental procedures. Scale bar represents 0.1 substitutions per site. (**B**) Pseudo-tetrameric structure of CNGK channels. Numbers 1 to 4, homologous repeats; S1 to S6, transmembrane segments; yellow cylinders, cyclic nucleotide-binding domain CNBD; asterisks, epitopes recognized by antibodies anti-repeat1 of *Dr*CNGK (polyclonal) and anti-repeat3 of *Dr*CNGK (YENT1E2, monoclonal). (**C**) Whole-cell recordings from zebrafish sperm at low (left upper panel) and high (middle panel) extracellular $K^+$ concentrations. Left lower panel: Voltage step protocol. Right panel: corresponding IV relations. (**D**) Whole-cell recordings from an isolated sperm head. Description see part C. (**E**) Whole-cell recording from zebrafish sperm (upper panel) and an isolated head (lower panel). (**F**) IV relation of recordings from part E. (**G**) Pooled IV relations ( ± sd) of currents from zebrafish sperm (filled circle, n = 23) and sperm heads (open squares, n = 6).

The following figure supplements are available for figure 1:

**Figure supplement 1.** Amino-acid sequence of the *Dr*CNGK channel.

**Figure supplement 2.** Separation of heads and flagella from whole sperm.

**Figure supplement 3.** Electrophysiological characterization of currents recorded from zebrafish sperm.

surfaces, sperm swim with their flagellum slightly inclined, which pushes the head against the wall and stabilizes sperm at the surface (*Denissenko et al., 2012*; *Elgeti et al., 2010*). Thus, fish sperm motility might be governed by specific hydrodynamic and haptic interactions with the egg surface and the micropyle.

Although the principal targets of a CNGK-mediated hyperpolarization – the $Na^+/H^+$ exchanger and CatSper – are absent in fish, vertebrate orthologues of the sperm CNGK channel are present in various fish genomes (*Figure 1A*). Here, we study the function of the CNGK channel in sperm of the freshwater fish *Danio rerio* (*Dr*CNGK). The *Dr*CNGK channel constitutes the principal $K^+$ channel in *D. rerio* sperm. Unexpectedly, cyclic nucleotides neither regulate CNGK channel activity nor sperm motility; instead, intracellular alkalization, a key mechanism to control sperm function in many species, strongly activates CNGK and, thereby, triggers a $Ca^{2+}$ signal and a motility response. Although its mechanism of activation is entirely different compared to sea urchin sperm, the principal CNGK function, namely to provide a hyperpolarization that triggers a $Ca^{2+}$ signal, is conserved. Our results show that sperm signalling among aquatic species shows unique variations that probably represent adaptations to vastly different ionic milieus and fertilization habits.

## Results

In several genomes, we identified genes encoding putative cyclic nucleotide-gated $K^+$ channels (CNGK) (*Figure 1A*). CNGK channels are primarily present in marine invertebrates, yet absent in the genomes of vertebrates such as birds, amphibians, and mammals, except freshwater fish and coelacanths. The CNGKs of freshwater fish appear to form a phylogenetic sub-group on their own (*Figure 1A*). Moreover, the CNGK channel exists in the unicellular choanoflagellate (*Salpingocea rosetta*), the closest living relative of animals (*Levin and King, 2013*; *Umen and Heitman, 2013*).

CNGK channels feature a chimeric structure. Their overall four-repeat pseudotetrameric architecture is reminiscent of voltage-dependent $Na_v$ and $Ca_v$ channels, whereas the pore carries the canonical GYG or GFG motif of $K^+$-selective channels (*Figure 1B* and *Figure 1—figure supplement 1*). Furthermore, CNGK channels are phylogenetic cousins of cyclic nucleotide-gated (CNG) channels and of hyperpolarization-activated and cyclic nucleotide-gated (HCN) channels (*Figure 1A*); each of the four repeats harbours a cyclic nucleotide-binding domain (CNBD) (*Figure 1B*). In fact, the CNGK channel of sea urchin sperm is activated at nanomolar cGMP concentrations (*Bönigk et al., 2009*).

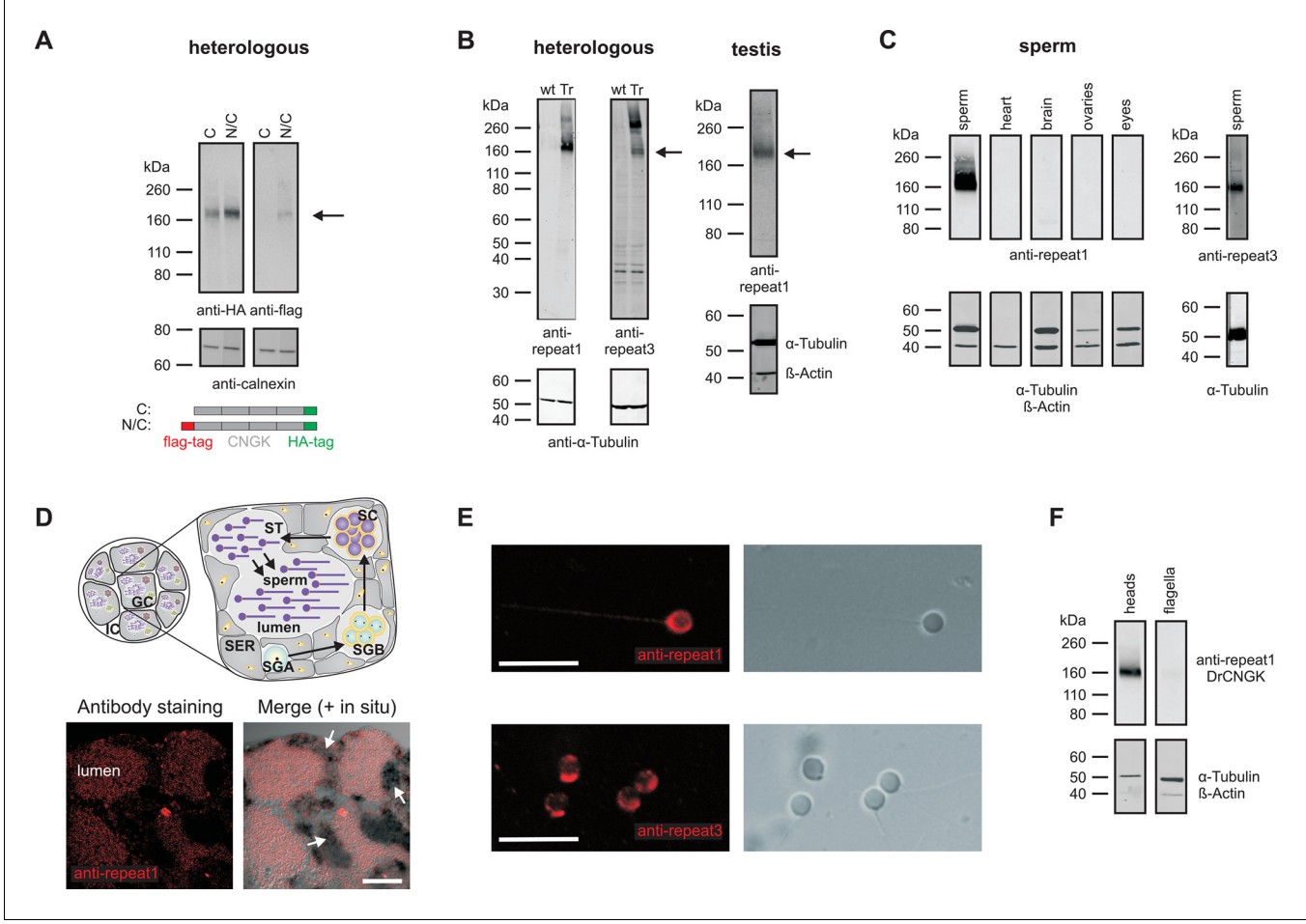

**Figure 2.** Localization of the *Dr*CNGK channel. (**A**) Western blot of membrane proteins (15 μg) from CHOK1 cells transfected with cDNA encoding either *Dr*CNGK with a C-terminal HA-tag alone (lane C) or with both, a C-terminal HA-tag and an N-terminal flag-tag (N/C). Apparent molecular weight $M_w$ is indicated on the left. (**B**) Characterization of anti-*Dr*CNGK antibodies. Left: Western blot of membrane proteins (10 μg) from HEK293 cells transfected with cDNA encoding *Dr*CNGK (Tr) and wild-type cells (wt). Right: Western blot of membrane proteins (15 μg) from zebrafish testis. (**C**) Western blot of membrane proteins (15 μg) from different zebrafish tissues. (**D**) Upper panel: Scheme of a testis cross-section. GC, germinal compartment; IC, intertubular compartment; SER, Sertoli cells; SGA, primary spermatogonia; SGB, secondary spermatogonia; SC, spermatocytes; ST, spermatids; scheme according to (*Nobrega et al., 2009*). Lower panel: Staining with anti-repeat1 antibody (red, left) and superposition (right) of the immunohistochemical image with a bright-field image of an in situ hybridization using an anti-*Dr*CNGK-specific RNA probe (arrows). Bar represents 50 μm. (**E**) Staining of zebrafish sperm with anti-repeat1 (upper left panel) and anti-repeat3 antibody (lower left panel). Bars represent 10 μm. The respective bright-field images are shown (upper and lower right panels). (**F**) Western blot of equal amounts of total membrane proteins (15 μg) from purified heads and purified flagella.

The following source data is available for figure 2:

**Source data 1.** Indicators of merit for the mass spectrometric results of a testis preparation.

**Source data 2.** Indicators of merit for the mass spectrometric results of different sperm preparations: whole sperm, head and flagella.

## Identification of a K⁺ current in sperm of *D. rerio*

We recorded currents from whole *D. rerio* sperm (*Figure 1—figure supplement 2*, left panel) in the whole-cell patch-clamp configuration. Voltage steps from a holding potential of -65 mV evoked slightly inwardly rectifying currents (*Figure 1C*). Two pieces of evidence established that currents are carried by K⁺ channels and not by Cl⁻ channels. The reversal potential ($V_{rev}$) shifted from $-77 \pm 3$ mV (n = 18) at 5.4 mM extracellular $[K^+]_o$ to $-7 \pm 1$ mV (n = 7) at 140 mM $[K^+]_o$ ($\triangle V_{rev} = 51 \pm 3$ mV/log $[K^+]$, n = 7) (*Figure 1C*, *Figure 1—figure supplement 3C*). Changing the intracellular $[Cl^-]$ did not

affect $V_{rev}$ (*Figure 1—figure supplement 3A,B*). These results demonstrate that the current is predominantly carried by K$^+$ ions. To localize the underlying K$^+$ channel, we recorded currents from isolated sperm heads (*Figure 1D-G*, *Figure 1—figure supplement 2*, middle panel). Head and whole-sperm currents displayed a similar K$^+$ dependence ($\triangle V_{rev} = 52 \pm 2$ mV/log [K$^+$], n = 6) (*Figure 1D*), rectification (*Figure 1F,G*), and amplitude (*Figure 1F,G*), suggesting that the underlying K$^+$ channel is primarily located in the head.

This result is unexpected, as ion channels involved in sperm signalling are usually localized to the flagellum. To test whether the *Dr*CNGK channel is also localized to the head, we used Western blot analysis and immunocytochemistry. To this end, the *Dr*CNGK protein was first characterized by heterologous expression in mammalian cell lines. *Dr*CNGK constructs with a C-terminal HA-tag or with two tags, a C-terminal HA-tag and an N-terminal flag-tag, were expressed in CHOK1 cells (*Figure 2A*). In Western blots, the anti-HA-tag and the anti-flag-tag antibody labelled proteins of the same apparent molecular mass ($M_w$) ($174 \pm 4$ kDa (n = 13) and $175 \pm 4$ kDa (n = 3), respectively) (*Figure 2A*). The $M_w$ is smaller than the predicted $M_w$ of 244.4 kDa. Because flag-tag and HA-tag antibodies recognized the N- and C-terminal end of the CNGK protein, respectively, we conclude that the 175-kDa band represents the full-length protein that, however, displays an abnormal electrophoretic mobility similar to other CNG channels (*Körschen et al., 1995*; *1999*).

We raised two antibodies against epitopes in repeat 1 and 3 of the *Dr*CNGK protein (*Figure 1B*, asterisks). Both antibodies labelled membrane proteins of about 170 kDa in Western blots of *Dr*CNGK-expressing CHOK1 cells, *D. rerio* testis, and sperm, but not of heart, brain, ovaries, and eyes (*Figure 2B,C*). To scrutinize the antibody specificity, we analyzed by mass spectrometry the ~170-kDa protein band from testis, mature whole sperm, isolated heads, and isolated flagella; 7, 23, 18, and 15 proteotypic *Dr*CNGK peptides were identified, respectively (*Figure 1—figure supplement 1*, *Figure 2—source data 1,2*). Peptides covered almost the entire polypeptide sequence (*Figure 1—figure supplement 1*). The presence of *Dr*CNGK in testis was confirmed by immunohistochemistry and in situ hybridization of *D. rerio* testis slices. The anti-repeat1 antibody labelled structures, most likely sperm, in the lumen of testicular compartments (*Figure 2D*, bottom left). An antisense RNA probe stained sperm precursor cells, in particular spermatocytes (*Figure 2D*, bottom right), but almost no primary or secondary spermatogonia.

Finally, the anti-repeat1 and anti-repeat3 antibodies intensely labelled the head and, to a lesser extent, the flagellum of single sperm cells (*Figure 2E*). In Western blots of isolated heads and flagella, the *Dr*CNGK was readily identified in head preparations, yet was barely detectable in flagella preparations (*Figure 2F*, n = 4). In summary, the CNGK channel is located primarily in the head of mature *D. rerio* sperm.

## The *Dr*CNGK channel is not sensitive to cyclic nucleotides

The sea urchin *Ap*CNGK channel is opened by cyclic nucleotides and mediates the chemoattractant-induced hyperpolarization (*Bönigk et al., 2009*; *Strünker et al., 2006*). Unexpectedly, patch-clamp recordings of K$^+$ currents from *D. rerio* sperm required no cyclic nucleotides in the pipette (*Figure 1C-G*). Therefore, we scrutinized the action of cyclic nucleotides on sperm K$^+$ currents. Mean current amplitudes were similar in controls and in the presence of either cAMP or cGMP (100 µM) in the pipette solution (*Figure 3A*). We used also caged cyclic nucleotides to study the K$^+$ current in the absence and presence of cyclic nucleotides in the same sperm cell (*Kaupp et al., 2003*). Photo-release of cAMP or cGMP from caged precursors did not affect K$^+$ currents, suggesting that cyclic nucleotides do not modulate *Dr*CNGK (*Figure 3B*). In contrast, the photo-release of cAMP or cGMP induced a rapid current increase in heterologously expressed cyclic nucleotide-gated channels *Ap*CNGK from sea urchin (*Figure 3—figure supplement 3A*). We also heterologously expressed the *Dr*CNGK channel in *X. laevis* oocytes. The current-voltage (IV) relation (*Figure 3C–F*), K$^+$ dependence (*Figure 3—figure supplement 1A-D*), and block by external TEA (*Figure 3—figure supplement 1E,F*) was similar to that of the K$^+$ current recorded from *D. rerio* sperm. Moreover, currents in oocytes were also insensitive to the membrane-permeable analogs 8Br-cAMP and 8Br-cGMP (*Figure 3C–F*), whereas perfusion with 8Br-cGMP increased currents in oocytes that express the *Ap*CNGK channel from *A. punctulata* sperm (*Figure 3—figure supplement 3B*).

Membrane-permeant caged cyclic nucleotides have successfully been used to study sperm motility in sea urchin (*Böhmer et al., 2005*; *Kashikar et al., 2012*; *Wood et al., 2005*) and humans (*Gakamsky et al., 2009*). We studied *D. rerio* sperm motility before and after photo-release of

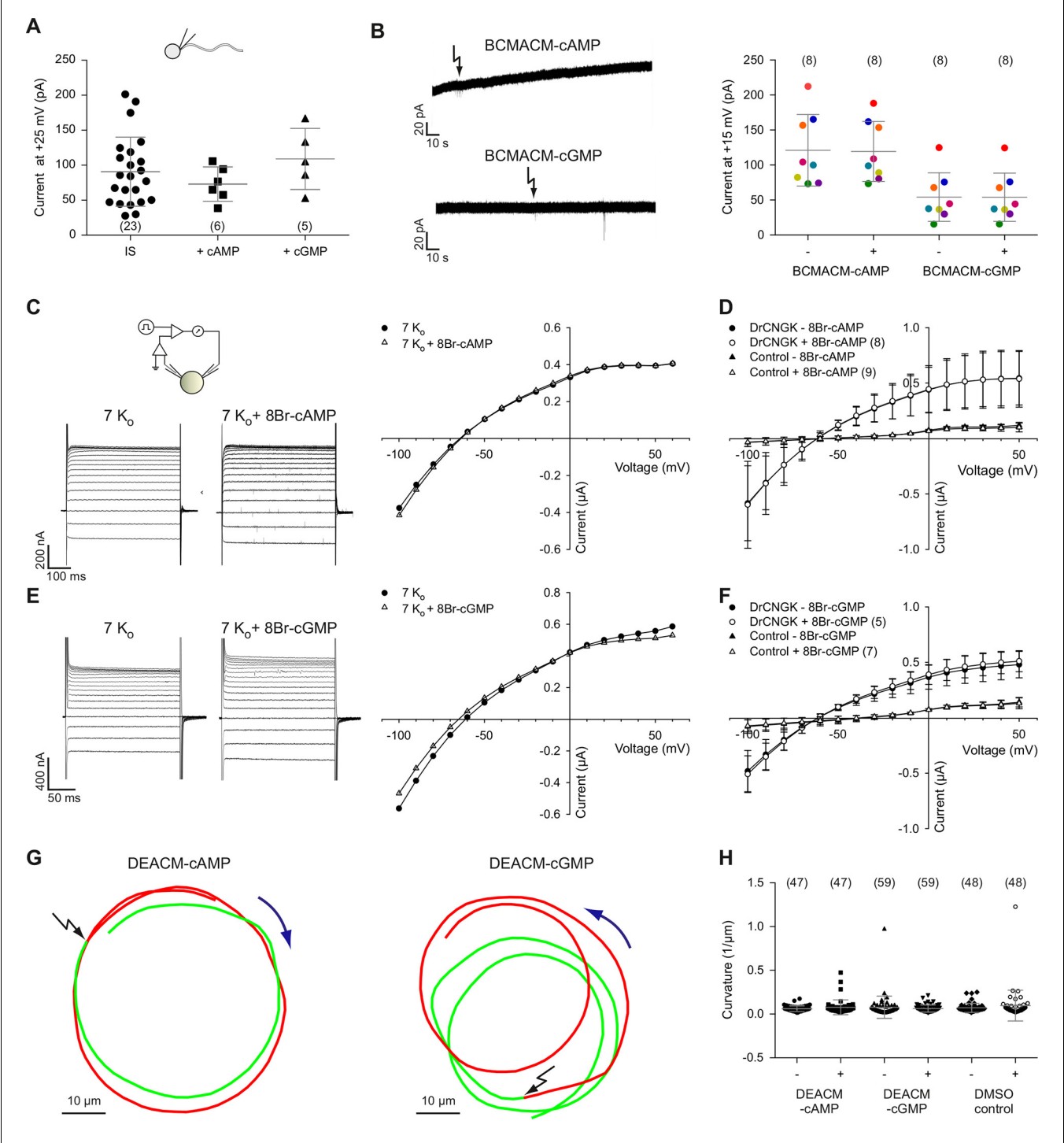

**Figure 3.** Cyclic nucleotides do not activate K⁺ channels in sperm. (A) Current amplitude of whole-cell recordings from zebrafish sperm at +25 mV in the absence or presence of 100 µM cAMP or cGMP in the pipette (control: 91 ± 49 pA (n = 23); cAMP: 73 ± 25 pA (n = 6); cGMP: 109 ± 44 pA (n = 5)). Individual data (symbols) and mean ± sd (gray bars), number of experiments in parentheses. (B) Photo-release of cyclic nucleotides from caged precursors inside sperm. Left panel: Whole-cell recordings at +15 mV from sperm loaded with 100 µM BCMACM-caged cAMP (upper panel) or BCMACM-caged cGMP (lower panel). Arrows indicate the delivery of the UV flash to release cyclic nucleotides by photolysis. Right panel: Mean current 3 s before (-) and 3 s after (+) the release of cAMP or cGMP. Statistics as in part A. Data points from individual sperm are indicated by identical colours. (C-F) Currents of heterologously expressed DrCNGK channels in the absence or presence of 8Br- analogs of cyclic nucleotides. (C) Left: Two-Electrode Voltage-Clamp recordings from DrCNGK-injected *Xenopus* oocytes. Currents shown are in the absence (left traces) and presence (right traces) of 10 mM 8Br-cAMP. Voltage steps as shown in *Figure 3—figure supplement 1A*. Right: IV relations of current recordings from the left panel. (D) Pooled IV

*Figure 3 continued*

curves from *Dr*CNGK injected and control oocytes; recordings in the absence and presence of 10 mM 8Br-cAMP. (E) Left: Two-Electrode Voltage-Clamp recordings from *Dr*CNGK injected *Xenopus* oocytes. Currents shown are in the absence (left traces) and presence (right traces) of 10 mM 8Br-cGMP. Right: IV relations of current recordings from the left panel. (F) Pooled IV curves from *Dr*CNGK-injected and control oocytes; recordings in the absence and presence of 10 mM 8Br-cGMP. (G) Swimming path before (green line) and after (red line) photo-release (black flash) of cAMP (left panel) or cGMP (right panel). The blue arrow indicates the swimming direction. Photo-release of cyclic nucleotides was verified by monitoring the increase of fluorescence of the caging group (*Figure 3—figure supplement 5*) (*Hagen et al., 2003*). (H) Path curvature before (-) and after (+) release of cAMP or cGMP. Sperm were loaded with 30 µM DEACM-caged cAMP or DEACM-caged cGMP. Statistics as in part A.

The following figure supplements are available for figure 3:

**Figure supplement 1.** $K^+$ dependence of heterologously expressed *Dr*CNGK channels in oocytes and channel block by tetraethylammonium (TEA).

**Figure supplement 2.** Sequence alignment of the individual CNBDs from the *Dr*CNGK and *Ap*CNGK channels.

**Figure supplement 3.** Photo-release of cyclic nucleotides in HEK cells expressing *Ap*CNGK channels and use of 8Br-analogs in *Ap*CNGK-injected oocytes.

**Figure supplement 4.** Photo-release of cyclic nucleotides (A) or $Ca^{2+}$ (B) in sperm.

**Figure supplement 5.** Control of loading and release of DEACM-cAMP in zebrafish sperm.

cAMP (*Figure 3G*, left) or cGMP (*Figure 3G*, right) from caged precursors. The photo-release was followed by the increase of fluorescence of the free coumaryl cage (*Figure 3—figure supplement 5*) (*Bönigk et al., 2009*). Swimming behaviour, i.e. path curvature (*Figure 3H*) and swimming speed (*Figure 3—figure supplement 4A*) were not altered by photo-release, showing that neither cAMP nor cGMP play a major role in the control of sperm motility. In conclusion, we observe no action of cyclic nucleotides on the *Dr*CNGK channel and on the swimming behaviour of *D. rerio* sperm.

## Rectification of CNGK channels in zebrafish and sea urchin sperm is different

We noticed a much stronger rectification of currents carried by sea urchin *Ap*CNGK compared to zebrafish *Dr*CNGK (*Figure 4A*) and, therefore, investigated the origin of this pronounced difference. In classical $K^+$ channels, block by intracellular $Mg^{2+}$ (*Matsuda et al., 1987*) or spermine (*Fakler et al., 1995*) produces inward rectification. However, neither $Mg^{2+}$ nor spermine affected *Ap*CNGK rectification (*Figure 4B*). Instead, intracellular $Na^+$ blocked outward currents in a strong voltage- and dose-dependent fashion (*Figure 4B,D*). In the absence of $Na^+$, the IV relation of *Ap*CNGK and *Dr*CNGK channels converged (*Figure 4A,B*). We searched the pore regions of CNGK channels for clues regarding the molecular basis of the $Na^+$ block. In three of the four *Ap*CNGK pore motifs, we identified a Thr residue that in most $K^+$ channels is replaced by a Val or Ile residue (*Figure 4C*). When these Thr residues were changed to Val, the strong rectification of the mutant *Ap*CNGK channel was lost and the IV relation became similar to that of the *Dr*CNGK channel (*Figure 4E*). We also tested the reverse construct, introducing Thr residues into the pore motif of *Dr*CNGK channels. For unknown reasons, the mutants did not form functional channels.

Of note, the Thr residues are absent in CNGKs of freshwater organisms yet present in seawater organisms except for the sponge *Aq*CNGK (*Figure 4C*), suggesting that the different CNGK pores represent adaptations to vastly different ionic milieus.

## The *Dr*CNGK channel is controlled by intracellular pH

Intracellular pH ($pH_i$) is an important factor controlling sperm motility in marine invertebrates and mammals (*Alavi and Cosson, 2005*; *Dziewulska and Domagala, 2013*; *Hirohashi et al., 2013*; *Lishko et al., 2010*; *Lishko and Kirichok, 2010*; *Nishigaki et al., 2014*; *Santi et al., 1998*; *Seifert et al., 2015*). Moreover, in mouse sperm, the Slo3 channels and the CatSper channels are exquisitely pH-sensitive (*Kirichok et al., 2006*; *Schreiber et al., 1998*; *Zeng et al., 2011*; *2013*; *Zhang et al., 2006a*; *2006b*). Therefore, we examined whether *Dr*CNGK is controlled by $pH_i$.

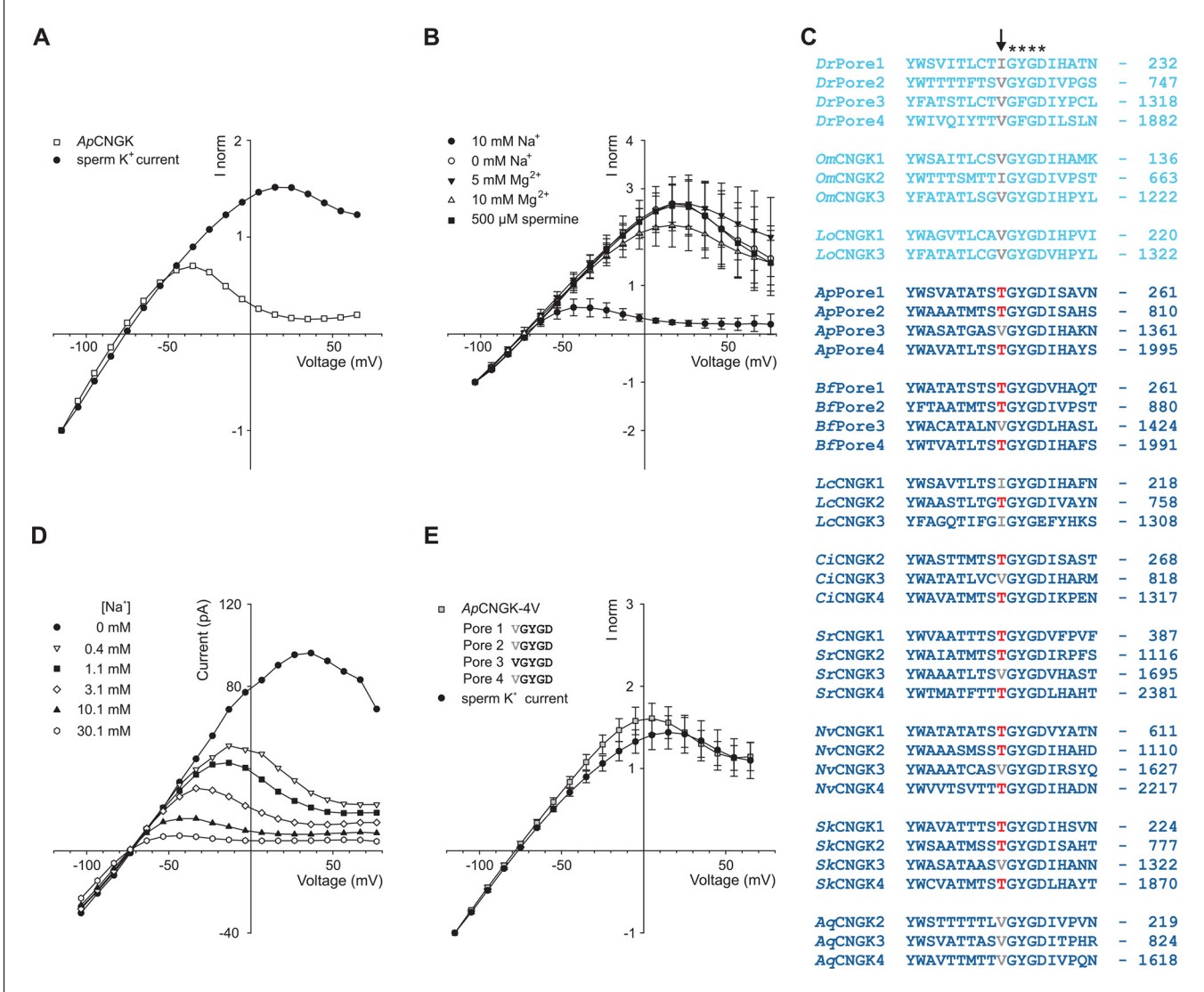

**Figure 4.** Comparison of sperm K$^+$ current with current from heterologously expressed $Ap$CNGK channels. (**A**) Normalized IV relations of whole-cell recordings from zebrafish sperm and $Ap$CNGK channels expressed in HEK293 cells. Pipette solution: standard IS. Bath solution: standard ES. Currents were normalized to -1 at -115 mV. (**B**) Normalized IV relations (mean current ± sd, n = 6) of inside-out recordings from $Ap$CNGK channels expressed in HEK293 cells. Pipette solution: standard ES, bath solution: NMDG-based IS with the indicated concentrations of Na$^+$, Mg$^{2+}$, and spermine. Currents were normalized to -1 at -103 mV. (**C**) Alignment of pore regions from different CNGK channels. Freshwater fishes are highlighted in light blue and seawater species in dark blue. The position of the last amino-acid residue is given on the right. Asterisks indicate the G(Y/F)GD selectivity motif. A key threonine residue that is conserved in three repeats of the $Ap$CNGK channel and other seawater species is highlighted in red (arrow). Hydrophobic amino acids at this position are indicated in gray. (**D**) IV relations of inside-out recordings of $Ap$CNGK channels expressed in HEK293 cells. Pipette solution: ES; bath solution: NMDG-based IS. Different Na$^+$ concentrations were added to the bath solution. (**E**) Normalized IV relations (mean current ± sd) of whole-cell recordings from zebrafish sperm (n = 18) and from $Ap$CNGK-4V channels (n = 7) expressed in HEK293 cells. Currents were normalized to -1 at -115 mV. Inset: amino-acid sequence of the pore region of the mutant $Ap$CNGK-4V. $Ap$CNGK channels were activated with 100 µM cGMP.

At pH$_i$ 6.4, almost no CNGK current was recorded from *D. rerio* sperm (**Figure 5A**, left panel, B,C). To rule out that an increase of Ca$^{2+}$ (from 91 pM to 7.7 nM, see Materials and methods) due to the reduced buffering capacity of EGTA at pH 6.4 is responsible for current inhibition, we recorded sperm currents at pH 7.4 with a free Ca$^{2+}$ concentration of 1 µM (**Figure 5—figure supplement 2**). Under these conditions, the K$^+$ current in sperm was still large, indicating that protons and not Ca$^{2+}$, at least for the concentrations tested, are responsible for current inhibition. Exposing sperm to 10 mM NH$_4$Cl rapidly elevates pH$_i$ (**Figure 5G**), because NH$_4$Cl overcomes the buffer capacity of HEPES

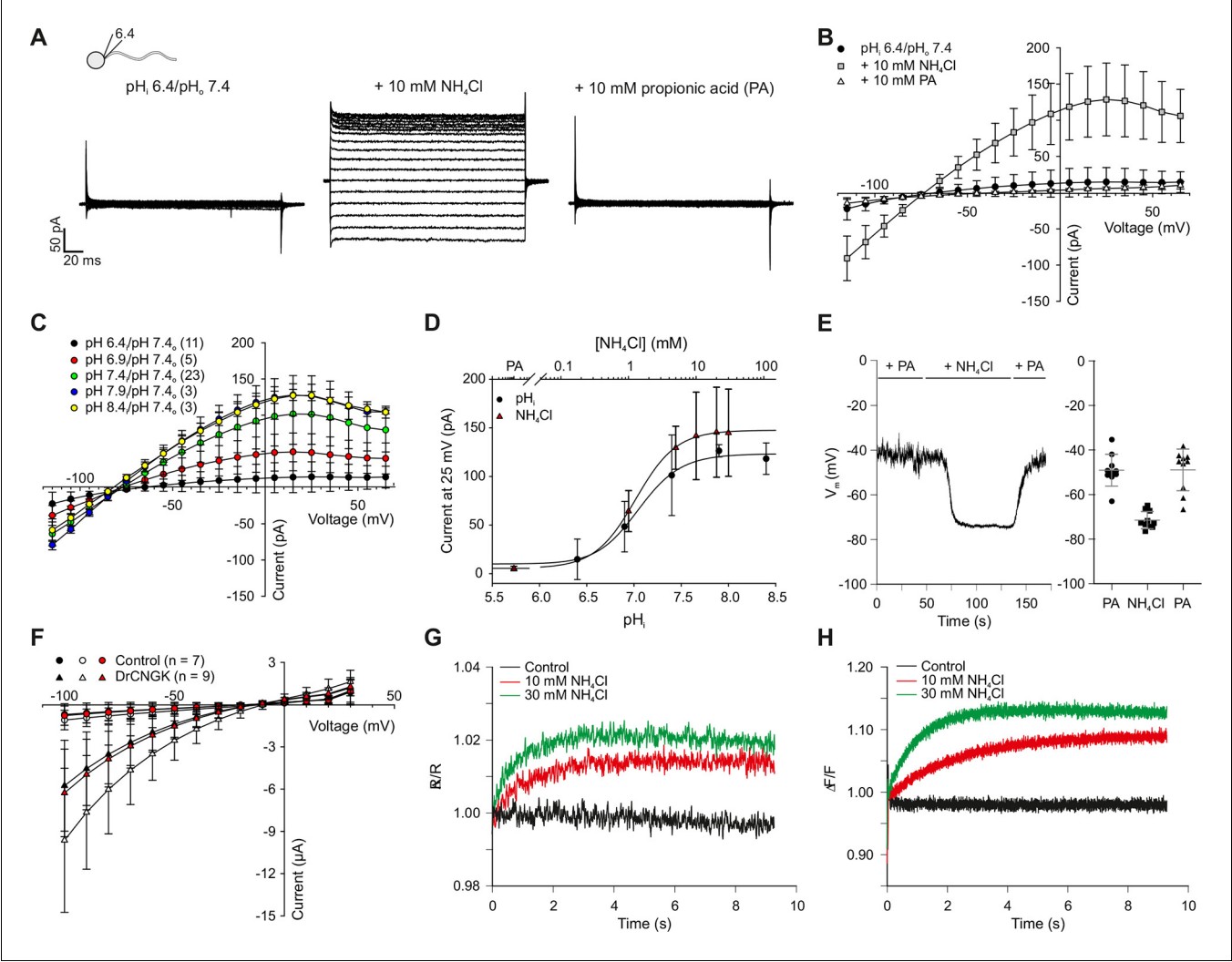

**Figure 5.** pH regulation of the *Dr*CNGK channel. (**A**) Whole-cell recordings from zebrafish sperm after perfusion with NH$_4$Cl or propionic acid. Voltage steps as shown in *Figure 1C*. Recordings at extracellular pH 7.4 and pipette pH 6.4 (left). NH$_4$Cl (10 mM, middle) or propionic acid (10 mM, right) was added to the bath. (**B**) Pooled IV curves for recordings from zebrafish sperm at a pipette pH$_i$ of 6.4 and in the presence of 10 mM NH$_4$Cl or 10 mM propionic acid (PA). (**C**) Pooled IV curves from recordings of zebrafish sperm at different intracellular pH$_i$. (**D**) Dependence of mean current ( ± sd) on intracellular pH$_i$ (circles, bottom axis) or in the presence of either 10 mM propionic acid (PA) or different NH$_4$Cl concentrations (triangles, top axis). (**E**) Recording of the voltage signal of zebrafish sperm in the current-clamp configuration. Pipette solution with an intracellular pH$_i$ of 6.4; recording in the presence of 10 mM NH$_4$Cl or 10 mM propionic acid (PA). Left panel: single recording. Right panel: individual data (symbols) and mean ± sd (gray bars), n = 10. (**F**) Pooled IV curves of Two-Electrode Voltage-Clamp recordings from heterologously expressed *Dr*CNGK channels and uninjected wild-type oocytes in 96 mM K$^+$ bicarbonate solution (black and red symbols) or 96 mM K$^+$ gluconate, including 1 mM NH$_4$Cl (white symbols, see *Figure 5—figure supplement 1* for recordings). (**G**) Changes in fluorescence of a zebrafish sperm population incubated with the pH indicator BCECF, recorded as the ratio of fluorescence at 549/15 nm and 494/20 nm (excited at 452/28 nm), before (black) and after the addition of 10 mM (red) or 30 mM (green) NH$_4$Cl. (**H**) Stimulation of sperm with NH$_4$Cl as in panel G using the Ca$^{2+}$ indicator Cal-520. Fluorescence was excited at 494/20 nm and recorded at 536/40 nm. Fluorescence F was normalized to the control value F$_0$ before stimulation.

The following figure supplements are available for figure 5:

**Figure supplement 1.** pH dependence of heterologously expressed *Dr*CNGK channels in oocytes.

**Figure supplement 2.** High intracellular Ca$^{2+}$ does not suppress *Dr*CNGK currents.

**Figure supplement 3.** Hypoosmotic conditions do not stimulate or diminish *Dr*CNGK currents in *Xenopus* oocytes.

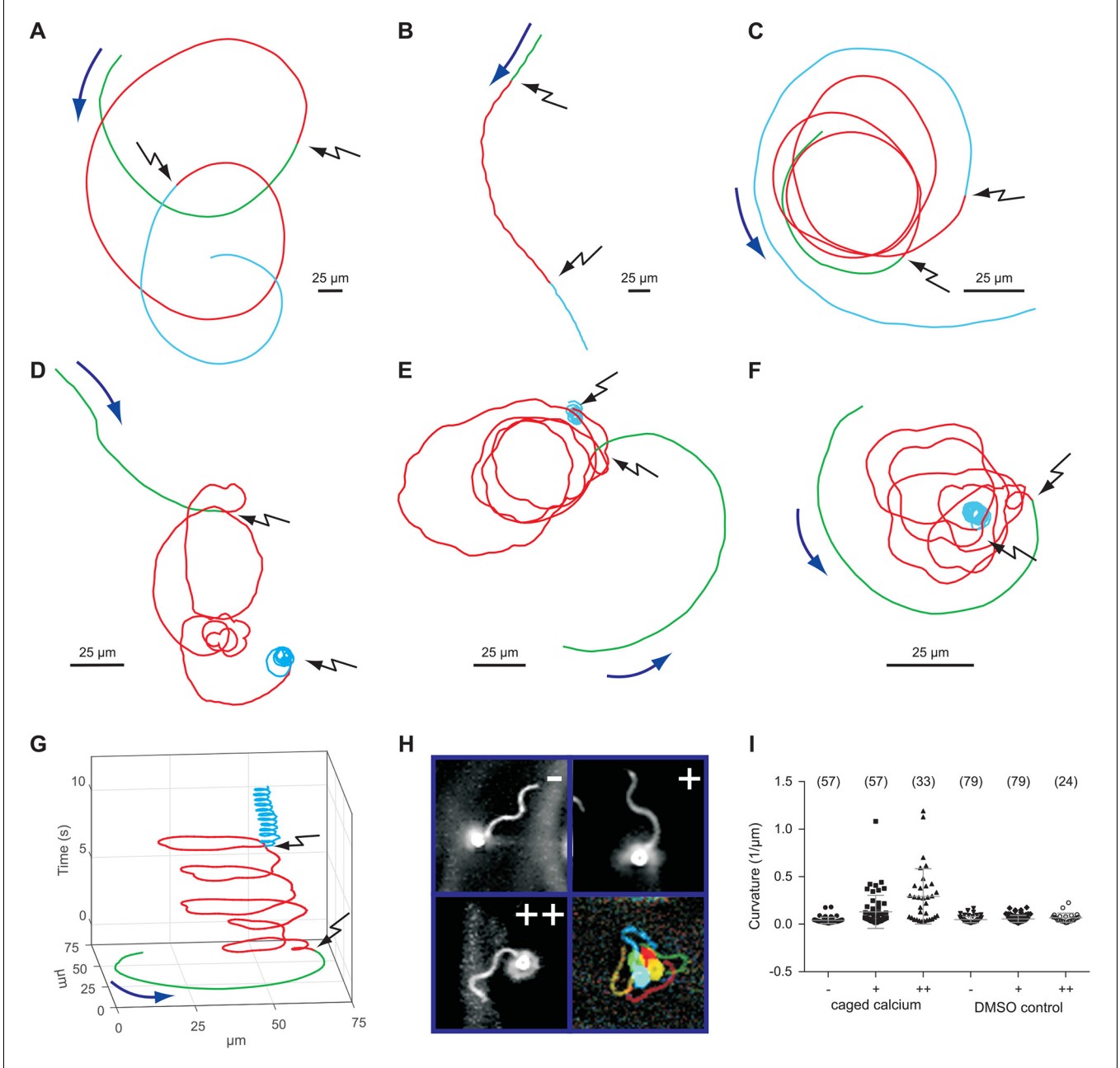

**Figure 6.** Sperm swimming behaviour upon $Ca^{2+}$ release. (A), (B), and (C) representative swimming paths of three different DMSO loaded sperm before and after application of UV light. (D), (E), and (F) representative averaged swimming paths of three different sperm before (green) and after $Ca^{2+}$ release by one (red) or two (cyan) consecutive UV flashes (black arrows). Curved blue arrows indicate the swimming direction of sperm. (G) Same swimming path shown in (F) including a temporal axis to facilitate the visualization of the changes in swimming path after consecutive flashes. Upon release (black arrows), the curvature of the swimming path progressively increases and the cell finally spins around the same position. (H) Representative flagellar shapes before (-), after $Ca^{2+}$ release by one (+) or two consecutive flashes (++), and during cell spinning against the wall (bottom right). Consecutive frames every 100 ms are shown in different colours. Sequence order: red, green, blue, and yellow. (I) Mean curvature before (-) and after one (+) or two (++) UV flashes. Individual data (symbols) and mean ± sd (gray bars), number of experiments in parentheses.

at pH 6.4 (*Boron and De Weer, 1976*; *Seifert et al., 2015*; *Strünker et al., 2011*). Alkaline $pH_i$ strongly enhanced CNGK currents (*Figure 5A*, middle panel, B,C). Subsequent superfusion with 10 mM propionic acid, which lowers $pH_i$, completely reversed the $NH_4Cl$-induced CNGK currents (*Figure 5A*, right panel, B and C). The $NH_4Cl$ action was very pronounced: 1 mM activated

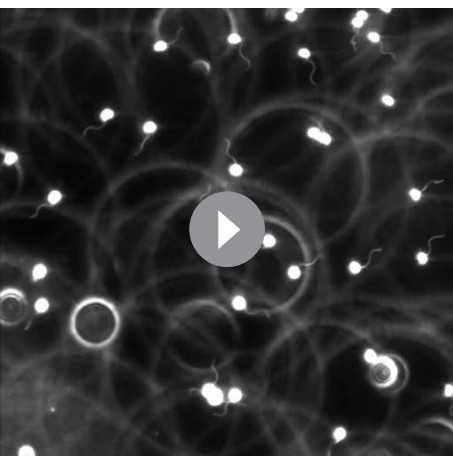

**Video 1.** Behavioural response of zebrafish sperm to successive $Ca^{2+}$ release. Representative recording of zebrafish sperm loaded with NP-EGTA (40 µM). Upon release of $Ca^{2+}$, the swimming path curvature increases and, eventually, sperm spin against the wall of the recording chamber with their flagellum pointing away from the wall. Video recorded using dark-field microscopy at 30 frames per second using a 20x magnification objective. The field of view corresponds to 410 µm. The Video is shown in real time.

approximately 40% of the CNGK current; at 10 mM, the current was maximal (*Figure 5D*, triangles). To quantitatively determine the pH dependence, we recorded sperm $K^+$ currents at different intracellular $pH_i$ values (*Figure 5C,D* circles). The current is half-maximally activated at pH 7.08. The pH dependence allows to calibrate the $NH_4Cl$ action by superposing the data of the two different experimental conditions (*Figure 5D*): For example, at an $NH_4Cl$ concentration of 1.5 mM, $pH_i$ in sperm will increase from 6.4 to 7.08 (*Figure 5D*). Under current-clamp conditions, alkalization of *D. rerio* sperm with 10 mM $NH_4Cl$ evoked a rapid and reversible hyperpolarization from -49 ± 7 mV to -71 ± 4 mV (*Figure 5E*, n = 10). We also studied the pH regulation of the *Dr*CNGK channel expressed in *Xenopus* oocytes. Oocytes were first perfused with a $K^+$ bicarbonate solution, followed by a $K^+$ gluconate-based solution including 1 mM $NH_4Cl$ (*Figure 5—figure supplement 1*). 1 mM $NH_4Cl$ reversibly increased *Dr*CNGK currents by approximately 71%, demonstrating regulation by $pH_i$ (*Figure 5F*). Higher concentrations of $NH_4Cl$ further increased the *Dr*CNGK current; however, these conditions also elicited significant currents in control oocytes, thus precluding quantitative analysis. Furthermore, we tested, whether DrCNGK channels in oocytes are activated by hypoosmotic conditions. Reducing the osmolarity by ~50% does not significantly change *Dr*CNGK currents in oocytes (*Figure 5—figure supplement 3*, n = 4). In sea urchin sperm, CNGK-mediated hyperpolarization leads to an increase of intracellular $Ca^{2+}$. Therefore, we tested, whether the $NH_4Cl$-induced hyperpolarization (*Figure 5E*) and alkalization (*Figure 5G*) evokes a $Ca^{2+}$ response. In fact, mixing of *D. rerio* sperm with 10 or 30 mM $NH_4Cl$ gave rise to a rapid $Ca^{2+}$ signal (*Figure 5H*, n = 4). The time course of the $pH_i$- and $Ca^{2+}$ signal was similar, suggesting that the CNGK-mediated hyperpolarization triggers a $Ca^{2+}$ influx. We conclude that *Dr*CNGK represents a pH-sensitive channel that is strongly activated at alkaline $pH_i$; the ensuing hyperpolarization, like in sea urchin sperm, produces a $Ca^{2+}$ signal.

## $Ca^{2+}$ controls swimming behaviour of *D. rerio* sperm

We studied the role of $Ca^{2+}$ for motility of *D. rerio* sperm using photo-release of $Ca^{2+}$ from caged $Ca^{2+}$ (NP-EGTA). Sperm motility was activated by hypoosmotic dilution (1:20 into 70 mM $Na^+$ ES, 167 mOsm x $L^{-1}$) and was followed under a dark-field microscope (*Figure 6*, *Video 1*). Unstimulated sperm swam on curvilinear trajectories of low curvature (*Figure 6A-F*, green segment, *Video 1*). A UV flash almost instantaneously increased path curvature in NP-EGTA-loaded sperm (*Figure 6D-F,I*), but not in control sperm (*Figure 6A-C,I*); sperm swam on much narrower arcs (*Figure 6D-F*, red segment, *Video 1*). The increase of path curvature was even more pronounced after a second UV flash (*Figure 6 D-F*, cyan segment, I, *Video 1*). Many cells were pushing against the wall of the observation chamber and performed a 'spinning' or 'drilling' behaviour, as if to penetrate the wall (*Figure 6G*). The asymmetry of the flagellar beat increased with each consecutive photo-release of $Ca^{2+}$ and eventually the flagellum pointed away from the glass surface (*Figure 6H*, *Video 1*). This swimming behaviour likely represents a strategy followed by sperm on its search for the micropyle, a small opening (outer diameter about 8 µm in diameter) (*Hart and Danovan, 1983*) on the surface of the much larger egg (about 0.75 mm in diameter) (*Selman et al., 1993*). A similar swimming behaviour during fertilization has been reported for sperm of herring and black flounder (*Cherr et al., 2008*; *Yanagimachi et al., 1992*; *2013*). We propose that the spinning or drilling movements observed after $Ca^{2+}$ release reflect the swimming behaviour in vivo down the narrow micropyle.

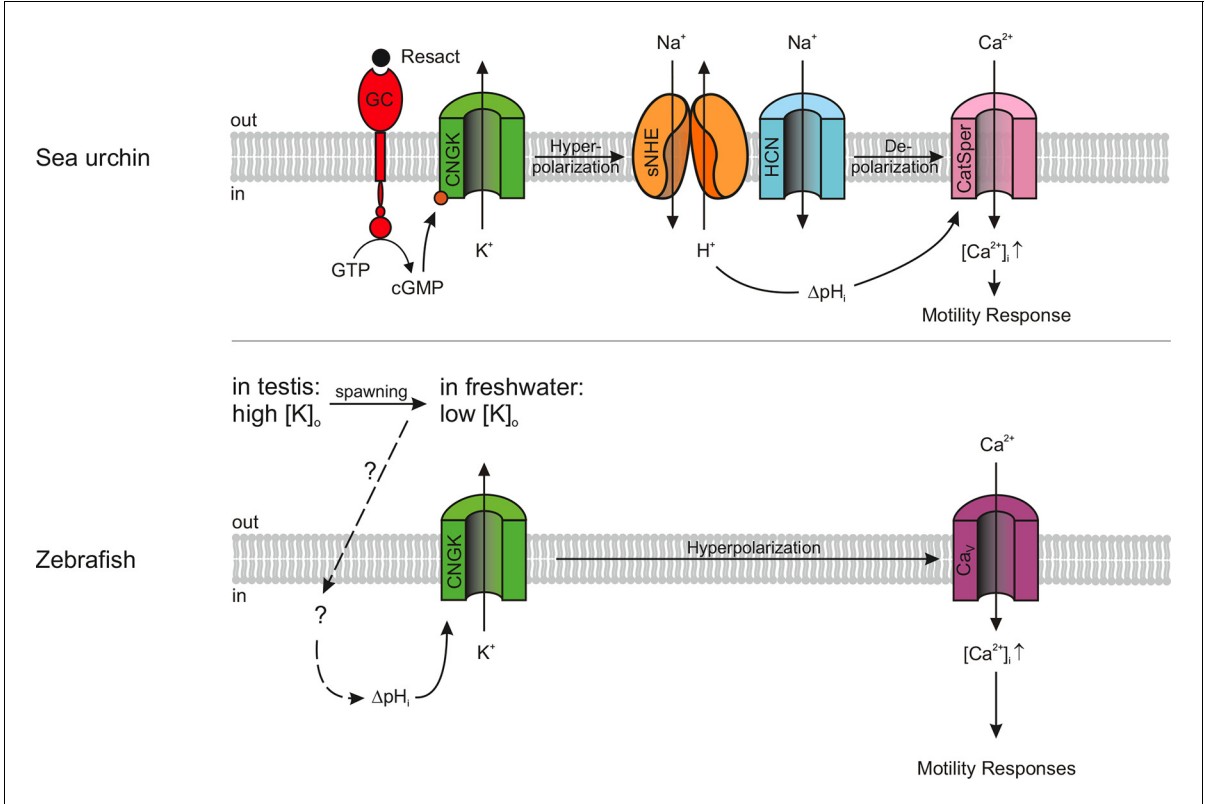

**Figure 7.** Models of signalling pathways in sea urchin and zebrafish sperm. Sea urchin (upper panel): Binding of the chemoattractant resact to a receptor guanylyl cyclase (GC) activates cGMP synthesis. Cyclic GMP opens $K^+$-selective CNG channels (CNGK), thereby, causing a hyperpolarization, which in turn activates a sperm-specific $Na^+/H^+$ exchanger (sNHE) that alkalizes the cell. Alkalization and subsequent depolarization by hyperpolarization-activated and cyclic nucleotide-gated (HCN) channels lead to the opening of sperm-specific CatSper channels. Zebrafish (lower panel): Upon spawning, $K^+$ efflux through CNGK hyperpolarizes sperm. An unknown mechanism of alkalization (dashed lines) modulates the open probability of CNGK channels; the ensuing hyperpolarization opens voltage-gated $Ca^{2+}$ channels ($Ca_v$).

## Discussion

A growing body of evidence reveals unexpected commonalities, but also notable differences among sperm from different species (for review (*Darszon et al., 2006*; *Kaupp et al., 2008*; *Yoshida and Yoshida, 2011*). Organisms, as phylogenetically distant as sea urchins and humans, share the CatSper channel as a common site of $Ca^{2+}$ entry into the sperm flagellum. By the same token, Slo3 $K^+$ channel orthologues in mouse and human sperm evolved different selectivity for intracellular ligands and might serve different functions (*Brenker et al., 2014*; *Chavez et al., 2013*; *Lishko et al., 2012*; *Santi et al., 2009*; *Zeng et al., 2013*). Here, we characterize a novel variant of CNGK channels in zebrafish sperm, whose key features depart from those of CNGK channels of marine invertebrates (*Figure 7*).

First, although the *D. rerio* CNGK carries four canonical CNBDs, it is gated by $pH_i$ rather than cyclic nucleotides, indicating that the CNBDs have lost their genuine ligand selectivity. The related *Ap*CNGK channel from sea urchin sperm is also unique in that it displays an unusual cGMP dependence: Unlike "classic" cooperative CNG channels, it is gated by binding of a single cGMP molecule to the third CNBD, implying that the other CNBDs are non-functional (*Bönigk et al., 2009*). Because the *Ap*CNGK channel is activated through binding of cGMP to the third repeat, we searched for sequence alterations in the third repeat of the *Dr*CNGK channel. Strikingly, in the C-linker region of the third repeat, we identified an insert of 42 amino-acid residues (*Figure 1—figure supplement 1*, blue) that is absent in other cyclic nucleotide-regulated channels. The insert shows no sequence similarity to any known functional domain of ion channels. One hypothesis is that this insert prevents the transmission of the binding signal to the channel pore. Another sequence peculiarity is identified in

the second repeat (*Figure 3—figure supplement 2*). At amino-acid position 934, an Ala residue replaces a highly conserved Arg residue that is crucial for cyclic-nucleotide binding (*Kaupp and Seifert, 2002*). The CNBDs of repeat 1 and 4 do not show obvious sequence abnormalities and could represent *bona fide* CNBDs (*Figure 3—figure supplement 2*). In recent years, many structures of CNBDs have been solved (*Clayton et al., 2004*; *Kesters et al., 2015*; *Kim et al., 2007*; *Rehmann et al., 2003*; *Schünke et al., 2011*; *Schünke et al., 2009*; *Zagotta et al., 2003*). Can we learn from these structures something about the *Dr*CNGK channel? Most of the CNBD structures feature a similar fold and are able to bind both cAMP and cGMP. However, for some CNG channels, ligands are full agonists, like cAMP and cGMP in the CNGA2 channel (*Dhallan et al., 1990*), only partial agonists, like cAMP in the CNGA1 channel (*Altenhofen et al., 1991*), or competitive antagonists, like cGMP in the bacterial SthK channel (*Brams et al., 2014*). Finally, in HCN channels, CNBDs interact and form a so-called gating ring (*Zagotta et al., 2003*), whereas in MloK1 channel, CNBDs do not interact at all (*Cukkemane et al., 2007*; *Schünke et al., 2009*; *2011*). In conclusion, at present, no sequence features can be identified that unequivocally explain the lack of cyclic-nucleotide regulation of the *Dr*CNGK channel.

The insensitivity of the *Dr*CNGK channel to cyclic nucleotides is, however, reminiscent of EAG and *h*ERG channels that carry classic CNBDs, yet are not gated by cyclic nucleotides (*Brelidze et al., 2009*; *2010*; *2012*). Instead, small molecules such as flavonoids have been suggested as ligands that bind to the CNBD and modulate channel activity (*Brelidze et al., 2010*; *Carlson et al., 2013*). Moreover, in the C-terminus of these CNBDs, a conserved segment of residues was identified that occupies the CNBD and serves as an intrinsic "ligand" (*Brelidze et al., 2012*; *Carlson et al., 2013*). We can only speculate, that, in addition to protons, as yet unidentified ligands might bind to and regulate the *Dr*CNGK channel. The apparent $pK_a$ value for channel activation by pH was approximately 7, suggesting that a His residue controls channel opening. There are a number of His residues in the C-linker of the four repeats that might serve as candidate sites. Future work is necessary to identify the site of pH regulation of the *Dr*CNGK channel.

To take on a new ligand selectivity or activation mechanism is also reminiscent of orthologues of the sperm-specific $K^+$ channel Slo3. Whereas the mouse Slo3 channel is exclusively controlled by $pH_i$ (*Brenker et al., 2014*; *Schreiber et al., 1998*; *Yang et al., 2011*; *Zeng et al., 2011*; *Zhang et al., 2006a*), the human Slo3 is primarily regulated by $Ca^{2+}$ (*Brenker et al., 2014*). In conclusion, the zebrafish CNGK is a striking example for a channel featuring a CNBD that is not gated by cyclic nucleotides. In general, CNBDs might represent sensor domains that can relay information on ligands other than cyclic nucleotides.

Second, signalling pathways that control sperm motility are located to the flagellum: The GC receptor for chemoattractant binding in sea urchin (*Bönigk et al., 2009*; *Pichlo et al., 2014*), the CatSper channel in humans, mice, and sea urchin (*Chung et al., 2014*; *Kirichok et al., 2006*; *Seifert et al., 2015*), the Slo3 $K^+$ channel in mice and humans (*Brenker et al., 2014*; *Navarro et al., 2007*), and the HCN and the CNGK channel in sea urchin (*Bönigk et al., 2009*; *Gauss et al., 1998*). In contrast, the *Dr*CNGK channel is located in the head rather than the flagellum. What might be the functional significance of such a peculiar location? The CNGK channel probably serves two related functions.

In seminal fluid, sperm of freshwater fish are immotile due to a high $[K^+]$ and high osmolarity. Upon release into hypoosmotic freshwater, sperm become motile for a few minutes (*Morisawa et al., 1983*; *Takai and Morisawa, 1995*; *Wilson-Leedy et al., 2009*). The osmolarity-induced activation hyperpolarizes sperm and induces a $Ca^{2+}$ signal (*Krasznai et al., 2000*). We propose that the CNGK triggers $Ca^{2+}$ signalling events upon spawning: In the high-$K^+$ seminal fluid, partially open CNGK channels keep sperm depolarized. When exposed to low-$K^+$ hypoosmolar conditions, sperm hyperpolarize and, ultimately, $Ca^{2+}$ is entering the cell and activates general motility (*Figure 7*).

Moreover, during the search for the micropyle on the egg surface, the sense of direction might be provided by haptic interaction with tethered molecules that line the opening or the funnel of the micropyle (*Iwamatsu et al., 1997*; *Ohta and Iwamatsu, 1983*; *Yanagimachi et al., 2013*). The haptic interactions could directly control CNGK activity in the head. For example, near or inside the micropyle, the CNGK might become further activated by alkaline pH and initiate the $Ca^{2+}$-dependent 'drilling' behaviour.

On a final note, the study of zebrafish sperm provides insight into adaptive mechanisms of sperm evolution. Boundary conditions might constrain sperm to develop different signalling strategies for similar functions. One obvious constraint is the ionic milieu, which strongly affects ion channel function. In freshwater, ion concentrations are low and opening of $Na^+$-, $K^+$-, and non-selective cation channels would hyperpolarize rather than depolarize cells. We speculate that, in freshwater fish, a depolarization-activated $Ca^{2+}$ channel like CatSper may not work and has been replaced by another $Ca_v$ channel.

Furthermore, the Thr/Val difference in *Ap*CNGK *versus* *Dr*CNGK, which determines $Na^+$ blockage, probably represents an adaptation to the respective ionic milieu. $Na^+$ blockage of sea urchin CNGK resists hyperpolarization in seawater and, thereby, facilitates the opening of depolarization-activated CatSper channels. The observation that CNGK channels from seawater organisms carry this Thr residue indicates a specific evolutionary pressure on this pore residue. Why is this Thr residue lost in CNGK channels of freshwater organisms? We speculate that $Na^+$ blockage disappeared along with the loss of CatSper genes and that $Ca^{2+}$ ions enter fish sperm through a $Ca^{2+}$ channel that is activated by hyperpolarization rather than depolarization. Future work needs to identify this $Ca^{2+}$ channel, its mechanism of activation, and its role for fertilization of teleost fish.

In summary, we identify a zebrafish CNGK channel that is activated at alkaline pH, and is set apart from its cousins of sea urchins that are activated by cGMP. Orthologues of CNGK also exist in the choanoflagellate *S. rosetta*, suggesting that this channel sub-family is phylogenetically ancient. Interestingly, this protozoon has a sexual life cycle: during anisogamous mating, small flagellated cells fuse with large cells (*Levin and King, 2013*). This mating behaviour represents the ancestor of sexual reproduction in animals (*Levin and King, 2013*; *Umen and Heitman, 2013*). The role of the *S. rosetta* CNGK channel for sexual reproduction without sperm will be interesting to study.

## Materials and methods

### Materials and reagents

Chemicals were purchased from AppliChem (Darmstadt, Germany), Biozym (Hessisch Oldendorf, Germany), Carl Roth (Karlsruhe, Germany), GE Healthcare Life Sciences (Munich, Germany), Life Technologies (Carlsbad, CA), Merck KgAa (Darmstadt, Germany), Merck Millipore (Billerica, MA), PolyScience (Warrington, PA), Qiagen (Hilden, Germany), Serva (Heidelberg, Germany), Sigma-Aldrich (Steinheim, Germany), and Thermo Scientifica (Waltham, MA). Enzymes and corresponding buffer solutions were ordered from Ambion (Austin, TX), MBI Fermentas (Vilnius, Lithuania), New England Biolabs (Frankfurt on the Main, Germany), and Roche (Basel, Switzerland). Primers were synthesized from Eurofins Genomics (Ebersberg, Germany). Chemicals for mammalian cell culture were ordered from Carl Roth, Life Technologies, and Biochrom (Berlin, Germany). CHOK1 and HEK293 cells were obtained from the American Type Culture Collection (ATCC, Manassas, VA).

### Cloning of the *Dr*CNGK gene

We identified two putative annotated sequences in the database: Eb934551 and XM_0013354. Eb934551 contained the putative N-terminal region and XM_0013354 contained the putative repeat 3, repeat 4, and parts of repeat 2. Using four sets of primer pairs, we did nested PCR reactions on testis cDNA to obtain the full-length sequence: the primer pairs #4811/#4812 and #4813/#4814 were used for XM_0013354 and the primer pairs C0274/C0275 and C0276/C0277 for Eb934551. The primer sequences were: TATTTCAAGTAGCTGTTACCG (#4811), ACATTCCCTTATAATAAT-GTCC (#4812), AAAAAAGCTAAGCTTTTCAGAAACACAG (#4813), AAAATCTGACAGGTACCCTG-CAGAATGC (#4814), CATACAGGATGCATGACCCC (C0274), CCAGGAATGTATGTGTAGGTC (C0275), GAGGAATTCATGCATGACCCCAGAGAAATGAAG (C0276) and CTCGGATCCGTATGTGT-AGGTCTTTAATTTCAGGG (C0277). Due to failure of expression, we used a codon-optimized version (human codon usage) of the *Dr*CNGK gene separated into three modules (Eurofins Genomics). Each module was flanked by restriction sites. The first module contained bases 1 to 2,108 and was flanked on the 5' end with BamHI and on the 3' end with XbaI; the second module contained bases 2,109 to 4,655 and was flanked on the 5' end with XbaI and on the 3' end with EcoRI; the third module contained bases 4,656 to 6,360 and was flanked on the 5' end with EcoRI and on the 3' end with NotI. At the 3' end, the coding sequence for the hemagglutinin tag (HA-tag) was added. The

construct was cloned into the pcDNA3.1 vector (Life technologies) (DrCNGK). To enhance expression levels, we added a QBI SP163 sequence (Stein et al., 1998) in front of the start codon (QBI-DrCNGK).

Moreover, we added the coding sequence for a flag-tag at the 5' end of the DrCNGK gene. We performed two PCR reactions with primer pairs C0991/C0962 and C0417/C0990 and a recombinant PCR reaction on the resulting PCR products with primer pair C0417/C0962. Primer sequences were: CCCGGACGGCCTCCGAAACCATGGACTACAAGGACGACGACGACAAGC (C0991), TTCAGAC-CGGCATTCCAAGCCC (C0962), CGCGGATCCAGCGCAGAGGCTTGGGGCAGC (C0417), GTCGT-CGTCGTCCTTGTAGTCCATGGTTTCGGAGGCCGTCCGGG (C0990).

ApCNGK pore mutants: for the pore mutant ApCNGK-4V, the following amino-acid substitutions in the ApCNGK wild-type channel (Bönigk et al., 2009) were produced: T252V, T801V and T1986V. Three PCR reactions were required for each mutation: two with the primers containing the point mutation and one recombinant PCR reaction. For the amino-acid exchange T252V, the primer pairs #4433/C0531 and C0530/#4409 were used and for the recombinant PCR, the primer pair #4433/#4409. The PCR product was cloned into the ApCNGK gene with restriction enzymes BamHI and XhoI. For the amino-acid exchange T802V, primer pairs #4436/C0533 and C0532/#4439 were used and for the recombinant PCR the primer pair #4436/#4439. The PCR product was cloned into the ApCNGK gene with restriction enzymes XhoI and XbaI. For the amino-acid exchange T1986V, the primer pairs #4412/C0535 and C0534/#4447 were used and for the recombinant PCR, the primer pair #4412/#4447. The PCR product was cloned into the ApCNGK gene with restriction enzymes BamHI and XbaI. The primer sequences were: GGTTCTGCTCGAGATTCTGTAGG (#4409), CAACAC-CGGATCCGGTGAGAGCAGTG (#4412), AAAGTTGGGATCCAATACAGCG (#4433), TACAGAATCT-CGAGCAGAACC (#4436), AAGTCTAGACGGTAGACTGATCGCCTGG (#4439), AAATCTAGATTAGGCATAATCGGGCACATCATAGGGATACACCACCGTTTGTCTCAGCG (#4447), GCCACCTCTGTAGGCTACGGAGAC (C0530), GTCTCCGTAGCCTACAGAGGTGGC (C0531), ATG-ACATCCGTGGGCTACGGAGAC (C0532), GTCTCCGTAGCCCACGGATGTCAT (C0533), CTGACCT-CCGTTGGCTACGGTGACATC (C0534), GTCACCGTAGCCAACGGAGGTCAGAG (C0535).

For expression in X. laevis oocytes, the DrCNGK and QBI-DrCNGK constructs were cloned into a modified version of the expression vector pGEMHEnew (Liman et al., 1992); because cloning of DrCNGK was only possible using BamHI and NotI, a NotI restriction site present in pGEMHEnew was removed and a new one was introduced into the multiple-cloning site. This new vector has been named pGEMHEnew-NotI. In vitro transcription to generate cRNA was performed using the mMES-SAGEmMACHINEKit (Ambion); the plasmid was linearized with SpeI.

For in situ hybridization, a short fragment of the DrCNGK gene coding for amino acids 1,085-1,219 was cloned into the pBluescript vector using PstI and HindIII. Nested PCR reactions were performed using for the 1st reaction the primer pair #4791/#4792 and for the 2nd reaction the primer pair #4793/#4794. Primer sequences were: ATTTTGCCGTGGGAGTCCATGG (#4791), AAGTCAATAT-TAAACGTTGCATCC (#4792), GGAAGCTTTCCGAAGCATTACAGCCG (#4793), GTTGGATCCAAG-TGTGTCACCCATGAC (#4794). For the antisense probe, the plasmid was linearized with HindIII and transcribed by T7 RNA polymerase. RNA was labeled with Digoxigenin (DIG RNA Labelling Mix, Roche).

## Preparation of testis, sperm, heads and flagella

Animals were sacrificed according to the "Guidelines for housing and care, transport, and euthanasia of laboratory fishes", ("Empfehlung für die Haltung, den Transport und das tierschutzgerechte Töten von Versuchsfischen", published by the Tierärztliche Vereinigung für Tierschutz e.V. January 2010). To obtain intact sperm, zebrafish male were anesthetized with MS-222 (0.5 mM, 3 min). After a brief wash with fresh water, the head was quickly separated from the body. The body of the fish was ventrally opened and two testis strands were removed and transferred into ES buffer (see Electrophysiology) or phosphate-buffered saline (PBS) containing (in mM: NaCl 137, KCl 2.7, $Na_2HPO_4$ 6.5, $KH_2PO_4$ 1.5, pH 7.4), additionally containing 1.3 mM EDTA, mPIC protease inhibitory cocktail (Sigma-Aldrich), and 1 mM DTT. Sperm were collected with a pipette tip. After 15 min, 80% of the supernatant was transferred into a fresh reaction tube. To separate heads from flagella, the sperm suspension was sheared 30-40 times on ice with a 24 gauge needle (Braun, Bethlehem, PA). The sheared suspension was centrifuged for 10 min (800xg, 4°C) to sediment intact sperm and sperm heads. This procedure was repeated twice. The purity of flagella preparations was assessed using

dark-field microscopy (*Figure 1—figure supplement 2*). For testis sections, males were ventrally sliced and kept overnight in 4% paraformaldehyde. After 24 hr, testis strands were removed and embedded in paraffin. Sections (8 μm) were made using a microtome (Leica, Wetzlar, Germany).

## Primary antibodies

A rabbit polyclonal antibody produced by Peptide Specialty Laboratories (PSL, Heidelberg, Germany) was directed against the cytosolic loop C-terminal of the CNBD of the first repeat (anti-repeat1, amino acids 483 - 497). Antibodies were purified with a peptide affinity column provided by PSL. Rat monoclonal antibody YENT1E2 (anti-repeat3) was directed against the extracellular loop between S5 and the pore region of the third repeat (amino acids 1,254 - 1,269). Anti-α-tubulin (mouse, B-5-1-2, Sigma-Aldrich), anti-β-actin (mouse, abcam, Cambridge, United Kingdom), anti-HA (rat, 3F10, Roche), anti-calnexin (rabbit, abcam), and anti-flag-tag (mouse, M2, Sigma-Aldrich) antibodies were used as controls.

## Immunocytochemistry, in situ hybridization, and Western blot analysis

Sperm were immobilized on SuperFrost Plus microscope slides (Thermo Fisher Scientific, Waltham, MA) and fixed for 5 min with 4% paraformaldehyde. After preincubation with 0.5% Triton X-100 and 5% chemiblocker (Merck Millipore) in PBS, sperm were incubated for 1 hr with antibodies YENT1E2 (1:10) or anti-repeat1 (1:500) diluted in 5% chemiblocker (Merck Millipore) and 0.5% TritonX-100 in PBS (pH 7.4). Sperm were visualized with Cy3-conjugated secondary antibody (Jackson ImmunoResearch Laboratories, West Grove, PA).

For in situ hybridization, tissue was permeabilized with protein kinase K (1 μg/ml in 0.1 M Tris/HCl, pH 8.0) and hybridized using the *Dr*CNGK-3 antisense probe. After washing, the antibody staining was performed using an anti-Digoxigenin antibody (1:500, Roche) conjugated with alkaline phosphatase. RNA was visualized with a mixture of nitro-blue tetrazolium chloride (500 μg) and 5-bromo-4-chloro-3'-indolyphosphate p-toluidine salt (188 μg, Roche). Cross sections were covered with a glass slip. For antibody staining of the in situ hybridization sections, cover slips were removed keeping the slides 5 min in xylene and briefly in PBS. This step was repeated; the fixative was removed from the sections. Afterwards, sections were stained as described for sperm immunocytochemistry. Proteins were probed with antibodies: anti-repeat1 (1:500) or anti-repeat3 (1:10) and visualized with Cy3-conjugated secondary antibody (Jackson ImmunoResearch Laboratories).

For Western blotting, zebrafish tissue or cells heterologously expressing the *Dr*CNGK channel were resuspended in PBS buffer containing 1.3 mM EDTA, mPIC protease inhibitor cocktail (Sigma-Aldrich), and 1 mM DTT. For lysis, cells were triturated 20x with a cannula (24G, Braun) and sonicated three times for 15 s. After a clearing spin (25,000xg, 30 min, 4°C), the pellet was resuspended and sonicated two times for 10 s in 200 mM NaCl, 50 mM Hepes (pH 7.5), mPIC, and 1 mM DTT. Triton X-100 was added to a final concentration of 1%. Proteins were solubilized for 1–2 hr at 4°C. A final clearing spin (10,000xg, 20 min, 4°C) was performed. For Western blot analysis, proteins were separated using 4-12% NuPAGE gradient gels (Life Technologies) and transferred overnight (4°C, 12–15 V) onto PVDF membranes (Immobilion FL, Merck Millipore), using a Xcell SureLock minigel chamber (Life Technologies). Membranes were incubated with Odyssey blocking buffer (LI-COR Biosciences, Lincoln, NE). Proteins were probed with the following antibodies: anti-repeat1 (1:1,000), anti-repeat3 (1:10), anti-α-tubulin (1:2,000), anti-β-actin (1:1,000), anti-HA (1:1,000), anti-calnexin (1:5,000), and anti-flag-tag (1:200). Proteins were visualized using IRDye800CW-conjugated secondary antibodies (1:10,000, LI-COR Biosciences), IRDye680-conjugated secondary antibodies (1:10,000, LI-COR Biosciences), or horseradish peroxidase-conjugated secondary antibodies (1:5,000, Jackson ImmunoResearch Laboratories). Visualization took place either with a chemiluminescence detection system (LAS-3000 Luminescent Image Analyzer, FUJIFILM, Life Science, Stamford, CT) or with fluorescent secondary antibodies (Odyssey infrared imaging system, Li-Cor Bioscience). The Novex Sharp pre-stained protein standard (Life Technologies) was used as molecular mass standard.

## Electrophysiology

We electrically recorded from intact zebrafish sperm and from isolated sperm heads using the patch-clamp technique in the whole-cell configuration. Recordings were accomplished within 4 hr after preparation. Seals between pipette and sperm were formed at the neck region in standard

extracellular solution (ES). The following pipette solutions were used: standard intracellular solution (IS) (in mM): NaCl 10, $K^+$ aspartate 130, $MgCl_2$ 2, EGTA 1, $Na_2ATP$ 2, and Hepes 10 at pH 8.4, 7.9, 7.4, 6.9, or 6.4 adjusted with KOH; $Cl^-$-based IS (in mM): NaCl 10, KCl 130, $MgCl_2$ 2, EGTA 1, $Na_2ATP$ 2, and Hepes 10 at pH 7.4 adjusted with KOH. The following bath solutions were used: standard ES (in mM): NaCl 140, KCl 5.4, $MgCl_2$ 1, $CaCl_2$ 1.8, glucose 10, and Hepes 5 at pH 7.4 adjusted with NaOH; for $K^+$-based ES solutions, the equivalent amount of $Na^+$ was replaced by $K^+$ (concentrations are indicated in the Figure legends). Calculations of the free $Ca^{2+}$ concentrations were carried out using the Maxchelator program (http://maxchelator.stanford.edu/webmaxc/webmaxcE.htm) assuming a residual $Ca^{2+}$ concentration in water of 1 μM. At pH 6.4, $[Ca^{2+}]_i$ was 7.7 nM and at pH 7.4, it was 91 pM. To obtain an intracellular solution with 1 μM free $Ca^{2+}$, 1 mM $CaCl_2$ was added to the IS solution at pH 7.4.

Caged compounds (100 μM BCMACM-caged cAMP or 100 μM BCMACM-caged cGMP) were added to the IS. The final concentration of DMSO was 0.1%. The compounds were photolyzed by a ~1 ms flash of ultraviolet light from a Xenon flash lamp (JML-C2; Rapp OptoElectronic, Wedel, Germany). The flash was passed through a BP295-395 nm filter (Rapp OptoElectronic) and delivered to the patch-clamp chamber in the microscope by a liquid light guide. Pipette resistance in IS/ES was between 11.5 and 15.0 MΩ. Voltages were corrected for liquid junction potentials.

For functional studies in *X. laevis* oocytes, 50 nl *Dr*CNGK RNA (0.3, 0.4, and 0.6 μg/μl) per oocyte were injected. Oocytes were purchased from EcoCyte Bioscience (Castrop-Rauxel, Germany) or prepared from dissected animals. Briefly, frogs were anesthetized with MS-222 (0.5%, 10-20 min), follicles were removed, opened with forceps and washed several times with ND96 solution. For defolliculation, oocytes were transferred for 1–2 hr (RT) into $Ca^{2+}$-free OR-2 solution containing 3 mg/ml collagenase type IV (Worthingthon Biochemical Corp., Lakewood, NJ). Defolliculated oocytes were stored in ND96 solution containing (in mM): NaCl 96, KCl 2, $MgCl_2$ 1, $CaCl_2$ 1.8, Hepes 10 at pH 7.6, pyruvate 2.5, and gentamycin 1. The OR-2 solution contained (in mM): NaCl 82.5, KCl 2.5, $MgCl_2$ 1, Hepes 5 at pH 7.6. We recorded in the Two-Electrode Voltage-Clamp configuration. Most data were recorded with a Dagan Clampator One (CA-1B, Dagan, Minneapolis, MN) amplifier and digitized with Digidata 1320A (Axon Instruments, Molecular Devices, Sunnyvale, CA). Analogue signals were sampled at 2 kHz. The holding potential was -80 mV. Pipette solution: 3 M KCl. Bath solutions were ND96-7K (in mM): NaCl 96, KCl 7, $MgCl_2$ 1, $CaCl_2$ 1.8, Hepes 10 at pH 7.4 adjusted with NaOH; $K^+$- based solution K96-7Na (in mM): NaCl 7, KCl 96, $MgCl_2$ 1, $CaCl_2$ 1.8, Hepes 10 at pH 7.4 adjusted with KOH. Recordings with reduced osmolarity were carried out in ND48-7K solution (in mM): NaCl 48, KCl 7, $CaCl_2$ 1.8, $MgCl_2$ 1, HEPES 10 at pH 7.4 adjusted with NaOH. Pipette resistance of voltage electrodes ranged between 1.5 and 3.0 MΩ and of current electrodes between 0.5 and 1.5 MΩ. Different analogues of cyclic nucleotides were added to the bath solution as indicated. Oocytes recordings with bicarbonate-based solutions were performed at Stanford University. Data were recorded with an OC-725C amplifier (Warner Instruments, Hamden, CT) using Patchmaster (HEKA Elektronik, Lambrecht, Germany) as acquisition software. Analogue signals were sampled at 1 kHz. The holding potential was -60 mV. Pipette solutions and pipette resistance as described above. Bath solutions: $K^+$ bicarbonate-based solution (in mM): NaCl 7, K-bicarbonate 96, $MgCl_2$ 1, $CaCl_2$ 1.8, Hepes 5 at pH 7.65. Solution was made fresh on each day of recording; $K^+$ gluconate-based solution (in mM): NaCl 7, K-gluconate 96, $MgCl_2$ 1, $CaCl_2$ 1.8, Hepes 5 at pH 7.65 adjusted with KOH. $NH_4Cl$ was dissolved in $K^+$gluconate-based solution.

We recorded *Ap*CNGK and mutant *Ap*CNGK currents from transfected (Lipofectamine 2000, Life technologies) HEK293 cells with the patch-clamp technique in the whole-cell configuration. A HEK293 cell line stably expressing the *Ap*CNGK channel was used for inside-out recordings. The pipette solution for whole-cell recordings was standard IS. Channels were activated with 100 μM cGMP. The pipette solution for inside-out recordings was standard ES. The following bath solutions were used for inside-out recordings: IS-30 NMDG-0Na$^+$ solution (in mM): NaCl 0, NMDG 30, KCl 110, EGTA 0.1, Hepes 10 at pH 7.4 adjusted with KOH; IS-NMDG-30Na$^+$ solution (in mM): NaCl 30, KCl 110, EGTA 0.1, Hepes 10 at pH 7.4 adjusted with KOH. 30 NMDG-0Na$^+$ and 0 NMDG-30Na$^+$ solutions were mixed to obtain the desired Na$^+$ concentrations. 100 μM Na$^+$-cGMP was added to the bath solution. For the solution with 0 mM Na$^+$, we used 100 μM Na$^+$-free cGMP. Pipette resistance in IS/ES was between 4.0 and 7.0 MΩ.

## Measurement of changes in intracellular Ca²⁺ concentration and pH

We measured changes in $[Ca^{2+}]_i$, and $pH_i$ in a rapid-mixing device (SFM-4000; BioLogic, Claix, France) in the stopped-flow mode using the $Ca^{2+}$ indicator Cal-520-AM (AAT Bioquest, Sunnyvale, CA) or the pH indicator BCECF-AM (Life Technologies). All sperm from a zebrafish male were diluted into 100 µl of ES solution and incubated with either 10 µM Cal-520-AM and 0.5% Pluronic for 120–180 min or 10 µM BCECF-AM for 10 min. Sperm were washed once, diluted 1:20 into ES solution, and loaded into the stopped-flow device. The sperm suspension was rapidly mixed 1:1 (vol/vol) with control ES solution or with ES solution containing $NH_4Cl$ to obtain final concentrations of 10 mM and 30 mM after mixing. Fluorescence was excited by a SpectraX Light Engine (Lumencor, Beaverton, OR). Cal-520 was excited with a 494/20 nm (Semrock, Rochester, NY), BCECF with a 452/45 nm (Semrock) excitation filter. Emission was recorded by photomultiplier modules (H9656-20; Hamamatsu Photonics). Fluorescence of Cal-520 was recorded using a 536/40 nm (Semrock) emission filter and normalized (without background subtraction) to the value before stimulation. BCECF fluorescence was recorded in the dual emission mode using a 494/20 nm (Semrock) and a 549/15 nm (Semrock) emission filter. The $pH_i$ signals represent the ratio of F494/549 and were normalized (without background subtraction) to the value before stimulation. All stopped-flow traces represent the average of 3–6 recordings. The signals were normalized to the first 5-10 data points before the onset of the signal to yield $\triangle F/F$ and $\triangle R/R$, respectively.

## Mass spectrometric identification of the *Dr*CNGK channel

Proteins of whole sperm, isolated heads, or flagella were resuspended in an SDS sample buffer and loaded on a SDS gel; after proteins had migrated approximately 1 cm into the separation gel, the gel was stained with Coomassie. The single gel band was excised for every sample, and proteins were in-gel digested with trypsin (Promega, Sunnyvale, CA); peptides were separated by RP-LC (180 min gradient 2–85% acetonitrile, (Thermo Fisher Scientific)) using a nanoAcquity LC System (Waters, Milford, MA) equipped with a HSS T3 analytical column (1.8 µm particle, 75 µm x 150 mm) (Waters) and analyzed twice by ESI-LC-MS/MS, using an LTQ Orbitrap Elite mass spectrometer (Thermo Fisher Scientific) with a 300-2,000 m/z survey scan at 240,000 resolution, and parallel CID of the 20 most intense precursors from most to least intense (top20) and from least to most intense (bottom20) with 60 s dynamic exclusion. All database searches were performed using SEQUEST and MS Amanda algorithm (*Dorfer et al., 2014*), embedded in Proteome Discoverer (Rev. 1.4, Thermo Electron 2008-2011, Thermo Fisher Scientific), with both a NCBI (26,623 entries, accessed December 20th, 2010) and a Uniprot (40,895 entries, accessed April 24th, 2014) zebrafish sequence protein database, both supplemented with the *Dr*CNGK protein sequence (*Figure 1—figure supplement 2*). Only peptides originating from protein cleavage after lysine and arginine with up to two missed cleavages were accepted. Oxidation of methionine was permitted as variable modification. The mass tolerance for precursor ions was set to 8 ppm; the mass tolerance for fragment ions was set to 0.6 amu. For filtering of search results and identification of *Dr*CNGK, a peptide FDR threshold of 0.01 (q-value) according to Percolator (*Käll et al., 2007*) two unique peptides per protein and peptides with search result rank 1 were required.

## Sequence analysis

Alignments for the calculation of the phylogenetic tree were done with ClustalOmega. Tree was depicted with Tree view (*Page, 1996*). The following ion channel sequences were used for the phylogenetic tree: CNGK channels from zebrafish (*Danio rerio, Dr*CNGK, XP_001335499.5); rainbow trout (*Oncorhynchus mykiss, Om*CNGK, CDQ79437.1); spotted gar (*Lepisosteus oculatus, Lo*CNGK, W5MTF2); West Indian ocean coelacanth (*Latimeria chalumnae, Lc*CNGK, H3BE11); sea urchin (*Arbacia punctulata, Ap*CNGK); acorn worm (*Saccoglossus kowalevskii, Sk*CNGK, XP_002731383.1); amphioxus (*Branchiostoma floridae, Bf*CNGK, XP_002592428.1); starlet sea anemone (*Nematostella vectensis, Nv*CNGK, XP_001627832); vasa tunicate (*Ciona intestinalis, Ci*CNGK, XP_002123955); sponge (*Amphimedon queenslandica, Aq*CNGK, I1G982); choanoflagellate (*Salpingoeca rosetta, Sr*CNGK, XP_004992545.1); murine HCN channel 1 (*Mus musculus, m*HCN1, NP_034538), 2 (*m*HCN2, NP_032252), 3 (*m*HCN3, NP_032253.1), and 4 (*m*HCN4, NP_001074661), and the HCN channel from sea urchin (*Strongylocentrotus purpuratus, Sp*HCN1, NP_999729); rat cyclic nucleotide-gated channels CNGA1 (*Rattus rattus, r*CNGA1, NP_445949), A2 (*r*CNGA2, NP_037060), A3

(rCNGA3, NP_445947.1), and A4 (rCNGA4, Q64359); the KCNH channels from fruit fly (*Drosophila melanogaster*, *Dm*EAG, AAA28495) and human (*Homo sapiens*, *h*ERG, BAA37096.1); murine voltage-gated Na$_v$ channels (*Mus musculus*, *m*Na$_v$ 1.1, NP_061203 and *m*Na$_v$ 1.6, NM_001077499.2); murine voltage-gated Ca$_v$ channels (*Mus musculus*, *m*Ca$_v$1.1, NP_055008, *m*Ca$_v$2.3, NP_033912.2, and *m*Ca$_v$3.1, NP_033913.2); and voltage-gated K$_v$ channels from fruit fly (*Drosophila melanogaster*, *Dm*Shaker, CAA29917.1) and mouse (*Mus musculus*, *m*K$_v$3.1, NM_001112739.1).

### Sperm motility

Sperm were loaded for 45 min at room temperature with either 30 µM DEACM-caged cAMP, 30 µM DEACM-caged cGMP, or 40 µM caged Ca$^{2+}$ (NP-EGTA, Life Technologies). A UV light-emitting diode (365-nm LED; M365L2-C, Thorlabs, Newton, NJ) was used for photolysis of caged compounds. Experiments using caged Ca$^{2+}$ and DEACM-caged nucleotides were carried out using a UV power of 25 and 22 mW, respectively. Flash duration was 300 ms. Pluronic (0.5%) was added to the incubation solution. Sperm were kept quiescent during incubation in ES solution (292 mOsm x L$^{-1}$). Swimming was initiated by a hypoosmotic shock diluting sperm 1:20 in an activation solution containing (in mM): NaCl 70, KCl 5.4, MgCl$_2$ 1, CaCl$_2$ 1.8, glucose 10, and Hepes 5 at pH 7.4 adjusted with NaOH (167 mOsm x L$^{-1}$). Swimming behaviour was observed with a dark-field condenser in an inverse microscope (IX71, Olympus, Tokio, Japan) with 20x magnification (UPLSAPO, NA 0.75). Movies were recorded at 30 Hz using a back-illuminated electron-multiplying charge-coupled device camera (DU-897D; Andor Technology, Belfast, United Kingdom). Sperm trajectories were tracked using custom-made software written in MATLAB (Mathworks). The software can be made available upon request. The average swimming path (ASP) was calculated by filtering the tracked coordinates with a second degree Savitzky-Golay filter with a 200 ms span. The curvature ($\kappa$) of the swimming path was calculated using the formula:bbb

$$\kappa = \left( \frac{\dot{x}\,\ddot{y} - \dot{y}\,\ddot{x}}{\dot{x}^2 + \dot{y}^2} \right)^{3/2}$$, where x and y are the coordinates of the ASP.

To assess cAMP loading and release, we recorded the fluorescence increase due to DEACM-OH release after photolysis of DEACM-caged cAMP (*Bönigk et al. 2009*). Release and fluorescence excitation was achieved simultaneously using the same UV LED (power 1.75 mW). Light was coupled to the microscope with a dichroic mirror (455DRLP, XF2034, Omega Optical, Brattleboro, VT) and fluorescence was long-pass filtered (460ALP; XF309; Omega Optical). Single cells were recorded at 50 Hz.

**Data analysis:** Statistical analysis and fitting of data were performed, unless otherwise stated, using Sigma Plot 11.0, GraphPadPrism 5, or Clampfit 10.2 (Molecular Devices). All data are given as mean ± standard deviation (number of experiments).

**Note:** All cell lines used in this study will be sent for STR profiling. Mycoplasma testing was performed using the Promokine Mycoplasma Test Kit 1/C (PromoCell GmbH, Heidelberg, Germany). Results of this test can be supplied upon request.

## Acknowledgements

We thank Drs. K Benndorf, and J Kusch (University Jena) and Drs. E Miranda-Laferte and P Hidalgo (Research Center Jülich), and Dr. M Goodman (Stanford University) for support with oocyte expression and H Krause for preparing the manuscript. SF was a fellow of the Boehringer Ingelheim Fonds. The authors declare no competing financial interests.

## Additional information

### Funding

| Funder | Grant reference number | Author |
| --- | --- | --- |
| Boehringer Ingelheim Fonds | Ph.D. Student Fellowship | Sylvia Fechner |

The funders had no role in study design, data collection and interpretation, or the decision to submit the work for publication.

## Author contributions

SF, Designed the project and experiments, Performed electrophysiology in mammalian cells, sperm and oocytes, biochemistry, cell biology, motility and stopped-flow experiments, Analysis and interpretation of data, Wrote, read and corrected the manuscript; LA, Performed motility experiments and analysis, Wrote, read and corrected the manuscript; WB, AM, Cloned the CNGK gene and mutants; TKB, Performed electrophysiology in sperm, Analysis and interpretation of data; RP, Performed motility experiments, Analysis and interpretation of data; CT, Performed the MS analysis; AP, Performed MS analysis; GS, Guided electrophysiology in oocytes; KRS, Performed in situ hybridization; EK, Produced the monoclonal antibodies; RS, Designed the project and experiments, Performed stopped-flow experiments, Performed electrophysiology in sperm, Analysis and interpretation of data, Wrote, read and corrected the manuscript; UBK, Designed the project and experiments, Wrote and corrected the manuscript, Analysis and interpretation of data, Drafting or revising the article

## Author ORCIDs

Gabriel Stölting, http://orcid.org/0000-0002-2339-0545

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
