## [Decision Letter]

Thank you for submitting your work entitled "A K^+^-selective CNG channel orchestrates Ca^2+^ signalling in zebrafish sperm" for peer review at *eLife*. Your submission has been favorably evaluated by Michael Marletta (Senior editor) and three reviewers, one of whom, Richard Aldrich is a member of our Board of Reviewing Editors.

The reviewers have discussed the reviews with one another and the Reviewing editor has drafted this decision to help you prepare a revised submission.

Summary:

The authors performed the first reported whole cell patch clamp recording of zebrafish sperm and uncovered an alkalinization-activated K^+^ channel. They also cloned and did preliminary characterization of an atypical K^+^ channel CNGK. Unlike mammalian sperm, the freshwater fish sperm do not have cyclic nucleotide activated-Ca^2+^ influx, nor do they have CatSper channels, which mediates the major Ca^2+^ flux in mammalian and sea urchin sperm. The data in the paper is consistent with the model in which the fresh water fish uses a different mechanism to mediate Ca^2+^ influx for sperm function: instead of using alkalinization-activated, depolarization-potentiated CatSper, the fish sperm use alkalinization-activated K^+^ channel followed by hyperpolarization-potentiated Ca^2+^ influx. The findings thus emphasize the different approaches animals have evolved for sperm Ca^2+^ signaling in response to the drastically different environment.

There are a fairly large number of required revisions. However many of them can be very easily addressed.

Essential revisions:

1) The conclusion that the alkalinization-activated K^+^ channel recorded from sperm using patch clamping is formed by CNGK needs to be more firmly established. The major evidence linking the two is from Figure 3 that show CNGK-injected *Xenopus* oocytes had K currents. The amplitudes of the currents shown in the figure are, however, quite small (<1 µA at -100 mV or +100 mV) and there is no control (e.g. H_2_O-injected oocytes) in the figure. Perhaps the most straightforward way to demonstrate that CNGK is responsible for the K^+^ current is to record from CNGK knockout sperm. This may take more than 3 months even with the CRISPR way to generate the KO, if the authors don't already have the mutant fish. At minimum, the authors should do a much more detailed comparison of the biophysical and pharmacological properties of the sperm K^+^ currents and currents generated by CNGK injection/transfection in *Xenopus* oocytes or mammalian cells.

2) Figure 3 – the points for current plus and minus cGMP seem to be exactly the same. Please clarify.

3) Figure 3 – What do the high values in the control distributions signify? It looks like there is something happening with the controls that is not in the treated points – please clarify.

4) A lot of the results are "no effect". Positive controls seem necessary for these.

5) Figure 1 – It may be more informative to show complete IV curves with error bars instead of the bar graphs at two different voltages?

6) Figure 1 – Are these redundant with Figure 1 left panels?

7) Figure 2 – If regular micrographs of the two images are available, that would be helpful. The images do not indicate how many sperms there were.

8) The CNG nucleotide insensitivity. There remains a formal possibility that there was a tightly bound endogenous ligand of some sort?

9) Figure 3—figure supplement 1 – A calibration bar is required here.

10) In the subsection “The *Dr*CNGK channel is not sensitive to cyclic nucleotides” – "We compared the sequences of *Dr*CNGK …". Perhaps citing a CNBG structure paper or two may be useful here? Does the structure 2K0G say anything interesting about *Dr*CNGK?

11) Figure 5 and pH manipulations. The internal pH manipulation experiments are somewhat complicated because the whole-cell internal solution was buffered to pH 6.4 with 10 HEPES. Presumably, 10 mM NH_4_Cl and propionic acid were so powerful that 10 mM HEPES was not sufficient. The authors could note this in the Results section (somewhere) to indicate this. It will be best to remove "pHi 6.4" from the middle and right panels from Figure 5 because the real pH value was probably not known? The authors could consider citing Boron and De Weer 1976 JGP for manipulating intracellular pH instead.

12) Figure 5, pH manipulations, and intracellular EGTA/Ca^2+^. One possible (unclear how probable) confounding factor in the pH manipulation experiments is that the internal solution calcium concentration was chelated with 1 mM EGTA, whose chealating action is very much dependent on pH. If the authors do not think this was a problem, they could add a note to indicate this?

13) If feasible at all, it would be more convincing to do inside out experiments so that the internal pH could be more directly manipulated.

14) Figure 5. Wouldn't it be more informative to show pooled IV curves with error bars?

15) In the subsection “Measurement of changes in intracellular Ca^2+^ concentration and pH” – Were the background signals subtracted? If so, how?

16) Figure 6 – Are the results from 3 different sperms?

17) Figure 6 and swimming. Some readers may want to see how "control" sperms (not loaded with caged molecules) respond to light flashes.

18) Figure 3 – "control oocytes". What are they? Uninjected? Water-injected? Perhaps IV curves from the "control" oocytes may help also.

19) Figure 4 – It is unclear how the IV curves were "normalized".

20) Figure 3 needs to include control from H_2_O-injected oocytes.

21) Most of the patch clamp recordings used "standard ES (in mM): NaCl 140, KCl 5.4, MgCl2 1, CaCl2 1.8, glucose 10, and Hepes 5 at pH 7.4 adjusted". This standard ES is very un-physiological for conditions that freshwater fish sperm face; channels recorded under such a condition may have properties very different from the native ones. Did the authors test bath with lower ionic strength?

22) Figure 3, as the pHi-sensitivity is a very essential component of the signaling model, the authors should include a better characterization of sensitivity. In addition to the use of NH_4_Cl, recordings with pipette solutions with varying pH should be used, perhaps similar to the characterization of Ksper in mouse sperm by Navarro et al. (Navarro et al. 2007, PNAS 104: 7688-7692). Figure 5 with NH_4_Cl dose response is a nice way to that the channel is pHi sensitive, but the responses need be more quantitative, unless the pH imaging in panels F & G can be calibrated to convert [NH_4_Cl] to pHi.

23) Figure 7, the model of zebrafish sperm Ca^2+^ signaling could be much strengthened if the authors could actually demonstrate the presence of CaV in the sperm with patch clamp recording. This might be hard if the current is small. However, Ca^2+^-permeable channels have been recorded in all the other sperm cells where patch clamping has been performed, including human and mouse sperm.

---

## [Author Response]

*Essential revisions:*

*1) The conclusion that the alkalinization-activated K^+^ channel recorded from sperm using patch clamping is formed by CNGK needs to be more firmly established. The major evidence linking the two is from Figure 3 that show CNGK-injected Xenopus oocytes had K currents. The amplitudes of the currents shown in the figure are, however, quite small (<1 µA at -100 mV or +100 mV) and there is no control (e.g. H_2_O-injected oocytes) in the figure. Perhaps the most straightforward way to demonstrate that CNGK is responsible for the K^+^ current is to record from CNGK knockout sperm. This may take more than 3 months even with the CRISPR way to generate the KO, if the authors don't already have the mutant fish. At minimum, the authors should do a much more detailed comparison of the biophysical and pharmacological properties of the sperm K^+^ currents and currents generated by CNGK injection/transfection in X. oocytes or mammalian cells.*

Unfortunately, no knockout sperm are available. Therefore, we characterized the K_+_ currents in sperm and oocytes in more detail. We have now determined quantitatively the pH-dependence of the K_+_ current in sperm. The apparent pH value of the sperm K_+_ current is ~7.08. In addition, we show that *Dr*CNGK-mediated currents in oocytes is also regulated by intracellular pH. These new results are now shown in Figure 5 and Figure 5 and Figure 5—figure supplement 1.

*2) Figure 3 – the points for current plus and minus cGMP seem to be exactly the same. Please clarify.*

We agree, the points seem to be identical; however, they are indeed different (see table below). We have now modified the plot such that each pair of recordings (before – after) is displayed in the same color.

**Table d36e7989:** 

**caged-cAMP**	**caged-cGMP**
before	after	before	after
73.19	73.50	29.96	30.20
156.82	153.68	124.90	124.44
212.30	188.31	37.56	37.15
165.23	161.90	15.62	15.73
104.31	108.84	36.52	36.45
82.43	89.13	67.90	67.68
74.29	80.26	44.79	44.36
99.86	98.89	75.67	75.96

*3) Figure 3 – What do the high values in the control distributions signify? It looks like there is something happening with the controls that is not in the treated points – please clarify.*

The illustration in fact is misleading. We originally plotted the relative change in current for injected and control oocytes. Because the currents in control oocytes are small, even small changes (for example of leak current) result in a large % change. We have now plotted the mean IV curves in the absence and presence of 8Br-cAMP and 8Br-cGMP for injected and control oocytes (new Figure 3). These complete IV curves are much more informative.

*4) A lot of the results are "no effect". Positive controls seem necessary for these.*

We observed “no effect” on *Dr*CNGK currents during perfusion of oocytes with 8Br-cAMP and 8Br-cGMP. As a positive control, we used the sea urchin *Ap*CNGK channel expressed in *Xenopus* oocytes. Perfusion with 8Br-cGMP consistently increased *Ap*CNGK currents (see below). We have included this experiment in Figure 3—figure supplement 3.

We also observed “no effect” including cyclic nucleotides in the pipette solution for patching zebrafish sperm. The same solutions activate heterologously expressed ApCNGK channels in HEK293 cells (see Figure 4). Furthermore, we tested the release of cAMP and cGMP from its precursor BCMACM for patch- clamp recordings in the whole-cell configuration. Here, we used heterologously expressed ApCNGK channels in HEK cells. Results are shown in Figure 3—figure supplement 3.

We also observed “no effects” releasing caged cyclic nucleotides in zebrafish sperm and studying motility. Here, we used a compound that increases its fluorescence when released. The increase in fluorescence was recorded during release. Therefore, we are confident that cyclic nucleotides are actually released. This data is now shown in Figure 3—figure supplement 5. Finally, the same compounds in *sea urchin* sperm and in *starfish* sperm consistently evoke behavior responses. Furthermore, release of caged calcium in zebrafish sperm evoke behavior responses, as shown in Figure 6.

*5) Figure 1 – It may be more informative to show complete IV curves with error bars instead of the bar graphs at two different voltages?*

Thanks for the suggestion. We have now plotted complete IV curves (see Figure 1)

*6) Figure 1 – Are these redundant with Figure 1 left panels?*

These are different experiments, yet recorded under the same conditions. In Figure 1, we display them on a larger scale to better illustrate the small currents recorded at physiological potassium concentrations. We are open to omit these panels, if requested.

*7) Figure 2 – If regular micrographs of the two images are available, that would be helpful. The images do not indicate how many sperms there were.*

We have now added the respective bright-field images to Figure 2.

*8) The CNG nucleotide insensitivity. There remains a formal possibility that there was a tightly bound endogenous ligand of some sort?*

We cannot rule out an endogenous ligand, but we can rule out that the CNGK is activated by cyclic nucleotides. For example, the EAG/HERG channel family bears a predicted cyclic nucleotide-binding domain that was suggested to bind ligands other than cyclic nucleotides, for example flavonoids. In addition, structures of these domains show that the CNBD binds a small C-terminal beta-strand, which can be considered as an intrinsic ligand (Brelitze et al. 2012). We mention this possibility in the Discussion.

*9) Figure 3—figure supplement 1 – A calibration bar is required here.*

Most likely, Figure 1—figure supplement 3 is meant. We have now included a calibration bar.

*10) In the subsection “The DrCNGK channel is not sensitive to cyclic nucleotides” – "We compared the sequences of DrCNGK…". Perhaps citing a CNBG structure paper or two may be useful here? Does the structure 2K0G say anything interesting about DrCNGK?*

We have now added a paragraph in the Discussion, where we discuss the sequences and refer to a number of structural papers about CNBDs.

*11) Figure 5 and pH manipulations. The internal pH manipulation experiments are somewhat complicated because the whole-cell internal solution was buffered to pH 6.4 with 10 HEPES. Presumably, 10 mM NH_4_Cl and propionic acid were so powerful that 10 mM HEPES was not sufficient. The authors could note this in the Results section (somewhere) to indicate this. It will be best to remove "pHi 6.4" from the middle and right panels from Figure 5 because the real pH value was probably not known? The authors could consider citing Boron and De Weer 1976 JGP for manipulating intracellular pH instead.*

We mention now in the Results section that NH_4_Cl/propionic acid overcome the buffer capacity of HEPES at pH 6.4. We changed the labels in the figures as suggested. Finally, we cite Boron & De Weer 1976.

*12) Figure 5, pH manipulations, and intracellular EGTA/Ca^2+^. One possible (unclear how probable) confounding factor in the pH manipulation experiments is that the internal solution calcium concentration was chelated with 1 mM EGTA, whose chealating action is very much dependent on pH. If the authors do not think this was a problem, they could add a note to indicate this?*

We have now estimated the free calcium concentration in the pipette solution, using the maxchelator program: (http://maxchelator.stanford.edu/webmaxc/webmaxcE.htm); at pH 6.4, [Ca_2+_]_i_was 7.7 nM and at pH 7.4, it was 91 pM. To estimate if the higher calcium level at pH 6.4 reduces the open probability of the channel, we have conducted recordings at pH 7.4 with a free calcium concentration of ~1 µM, much higher than the 7.7 nM at pH 6.4. Under these conditions, the sperm K_+_ current was still present. We have now included these results in (Figure 5—figure supplement 2).

*13) If feasible at all, it would be more convincing to do inside out experiments so that the internal pH could be more directly manipulated.*

Two members of our research group have extensively tried to record *Dr*CNGK currents from inside-out patches from *Xenopus* oocytes, without success (>200 inside-out patches, includingmacropatches). Either the channel does not survive patch excision or the density of channels in the membrane is too low.

*14) Figure 5. Wouldn't it be more informative to show pooled IV curves with error bars?*

We have replaced Figure 5 by pooled IV curves with error bars.

*15) In the subsection “Measurement of changes in intracellular Ca^2+^ concentration and pH” – Were the background signals subtracted? If so, how?*

For calcium recordings, fluorometric data was normalized to the level before stimulation. No background was subtracted. For pH recordings, emission was recorded at 494 nm and 549 nm. Without background subtraction, the ratio of F494/F549 was plotted. We have now included this information in the Materials and methods section.

*16) Figure 6 – Are the results from 3 different sperms?*

Yes, these are experiments from three different sperm. We have now included this information in the legend.

*17) Figure 6 and swimming. Some readers may want to see how "control" sperms (not loaded with caged molecules) respond to light flashes.*

We have now included recordings from three different “control” sperm in the new Figure 6.

*18) Figure 3 – "control oocytes". What are they? Uninjected? Water-injected? Perhaps IV curves from the "control" oocytes may help also.*

We have now included IV curves from uninjected oocytes in the absence and presence of 8Br- cAMP and 8Br-cGMP in Figure 3 (now Figure 3).

In addition, we added more controls and analysis of the reversal potential of uninjected and *Dr*CNGK-injected oocytes in the presence of different extracellular K_+_ concentrations (Figure 3—figure supplement 1).

*19) Figure 4 – It is unclear how the IV curves were "normalized".*

In Figure 4, currents were normalized to -1 at -115 mV; in Figure 4 currents were normalized to -1 at -103 mV. We have now included this information in the figure legend.

*20) Figure 3 needs to include control from H_2_O-injected oocytes.*

We have now included the data from uninjected oocytes in the new Figure 3.

*21) Most of the patch clamp recordings used "standard ES (in mM): NaCl 140, KCl 5.4, MgCl2 1, CaCl2 1.8, glucose 10, and Hepes 5 at pH 7.4 adjusted". This standard ES is very un-physiological for conditions that freshwater fish sperm face; channels recorded under such a condition may have properties very different from the native ones. Did the authors test bath with lower ionic strength?*

We have attempted to record from sperm while changing to low-osmolarity solutions. Unfortunately, we always lost the recording configuration during this procedure.

*22) Figure 3, as the pHi-sensitivity is a very essential component of the signaling model, the authors should include a better characterization of sensitivity. In addition to the use of NH_4_Cl, recordings with pipette solutions with varying pH should be used, perhaps similar to the characterization of Ksper in mouse sperm by Navarro et al. (Navarro et al. 2007, PNAS 104: 7688-7692). Figure 5 with NH_4_Cl dose response is a nice way to that the channel is pHi sensitive, but the responses need be more quantitative, unless the pH imaging in panels F & G can be calibrated to convert [NH_4_Cl] to pHi.*

We have now recorded from sperm at various well-defined intracellular pH values. Results and analysis have now been included in the new Figure 5. This result can be used as a “calibration” for the NH_4_Cl action.

*23) Figure 7, the model of zebrafish sperm Ca^2+^ signaling could be much strengthened if the authors could actually demonstrate the presence of CaV in the sperm with patch clamp recording. This might be hard if the current is small. However, Ca^2+^-permeable channels have been recorded in all the other sperm cells where patch clamping has been performed, including human and mouse sperm.*

Our results on other channels in zebrafish sperm are still preliminary. We have not observed classical voltage-activated Ca_2+_ channels. There is some evidence of a cation channel that is activated by hyperpolarization. However, to date we don’t know, if it is permeable to Ca_2+_ ions, let alone its molecular identity. Therefore, we rather prefer not to present any of these data.